# Consent in Crisis:
# The Rapid Decline of the AI Data Commons

**Shayne Longpre**[1], **Robert Mahari**[1], **Ariel Lee**[1], **Campbell Lund**[1], **Hamidah Oderinwale**[2], **William Brannon**[2], **Nayan Saxena**[2], **Naana Obeng-Marnu**[2], **Tobin South**[2], **Cole Hunter**[2], **Kevin Klyman**[2], **Christopher Klamm**[2], **Hailey Schoelkopf**[2], **Nikhil Singh**[2], **Manuel Cherep**[2], **Ahmad Mustafa Anis**[3], **An Dinh**[3], **Caroline Chitongo**[3], **Da Yin**[3], **Damien Sileo**[3], **Deividas Mataciunas**[3], **Diganta Misra**[3], **Emad Alghamdi**[3], **Enrico Shippole**[3], **Jianguo Zhang**[3], **Joanna Materzynska**[3], **Kun Qian**[3], **Kush Tiwary**[3], **Lester Miranda**[3], **Manan Dey**[3], **Minnie Liang**[3], **Mohammed Hamdy**[3], **Niklas Muennighoff**[3], **Seonghyeon Ye**[3], **Seungone Kim**[3], **Shrestha Mohanty**[3], **Vipul Gupta**[3], **Vivek Sharma**[3], **Vu Minh Chien**[3], **Xuhui Zhou**[3], **Yizhi Li**[3], **Caiming Xiong**[4], **Luis Villa**[4], **Stella Biderman**[4], **Hanlin Li**[4], **Daphne Ippolito**[4], **Sara Hooker**[4], **Jad Kabbara**[4], and **Sandy Pentland**[4]

[1]Team Leads   [2]Top Contributors   [3]Contributors (alphabetized)   [4]Advisors

**Data Provenance Initiative**
Correspondence: data.provenance.init@gmail.com

## Abstract

General-purpose artificial intelligence (AI) systems are built on massive swathes of public web data, assembled into corpora such as C4, RefinedWeb, and Dolma. To our knowledge, we conduct the first, large-scale, longitudinal audit of the consent protocols for the web domains *underlying* AI training corpora. Our audit of $14,000$ web domains provides an expansive view of crawlable web data and how codified data use preferences are changing over time. We observe a proliferation of AI-specific clauses to limit use, acute differences in restrictions on AI developers, as well as general inconsistencies between websites' expressed intentions in their Terms of Service and their robots.txt. We diagnose these as symptoms of ineffective web protocols, not designed to cope with the widespread re-purposing of the internet for AI. Our longitudinal analyses show that in a single year (2023-2024) there has been a rapid crescendo of data restrictions from web sources, rendering ~5%+ of all tokens in C4, or 28%+ of the most actively maintained, critical sources in C4, fully restricted from use. For Terms of Service crawling restrictions, a full 45% of C4 is now restricted. If respected or enforced, these restrictions are rapidly biasing the diversity, freshness, and scaling laws for general-purpose AI systems. We hope to illustrate the emerging crises in data consent, for both developers and creators. The foreclosure of much of the open web will impact not only commercial AI, but also non-commercial AI and academic research.

## 1 Introduction

The web has become the primary communal source of data, or "data commons", for general-purpose and multi-modal AI systems. The scale and heterogeneity of web-sourced training datasets provide the foundation for both open and closed AI systems, such as OLMo [1], GPT-4o [2], and Gemini [3]. However, the use of web content for AI poses ethical and legal challenges to data consent, attribution, copyright, and the potential impact on creative industries [4, 5, 6, 7]. This has spurred new initiatives to better verify data quality and provenance [8, 9, 10, 11, 12], isolate public domain and permissively

38th Conference on Neural Information Processing Systems (NeurIPS 2024) Track on Datasets and Benchmarks.

licensed data [13], and integrate new infrastructure to signal [14], detect [15], and even evade the use of data for AI training [16].

The focus of this work is to understand the evolving role of the internet as a primary ingredient to AI, and how AI has collided with the limited protocols that govern data use. Web data is traditionally collected using *web crawlers*—automatic bots that systematically explore the internet and record what they see. However, the mechanisms for indicating restrictions to web crawlers, such as the Robots Exclusion Protocol (REP), were not designed with AI in mind [17]. (The REP is referred to as robots.txt in practice.) As such, we examine their (in)ability to communicate the nuances in how content creators wish their work to be used, if at all, for AI. And more broadly, we analyze how AI is already re-shaping the culture of web consent, and how this is shifting the landscape for AI training data. Our results foretell significant changes not only to AI data collection practices and data scaling laws, but also the structure of consent on the open web, which will impact more than AI developers.

To this end, we present a large-scale audit of the web sources underlying three open AI training corpora: C4 [18], RefinedWeb [19], and Dolma [20]. In contrast to prior audits that assess datasets—curated snapshots of data—this work looks *beneath* the datasets at the web domains they were derived from, and traces the temporal evolution of these sources. We are, to our knowledge, the first to systematically measure detailed provenance, crawler consent mechanisms, and content monetization factors, all relevant to the responsible downstream use of this data. These analyses enable us to trace fundamental distribution shifts in how preference signals are expressed and the inadequacy of existing tools. Our work has several key findings:

1. **A proliferation of restrictions on the AI data commons.** We find a rapid proliferation of restrictions on web crawlers associated with AI development in both websites' robots.txt and Terms of Service. We estimate that in one year (Apr 2023 to Apr 2024), ~25%+ of tokens from the domains most critical to model training, and ~5%+ of tokens from the entire corpora of C4, RefinedWeb, and Dolma have since become restricted by robots.txt. Forecasting these trends forward shows a decline in unrestricted, open web data year-over-year.
2. **Consent asymmetries & inconsistencies.** OpenAI's crawlers are significantly more restricted than those of other AI developers. More broadly, preference signaling mechanisms like robots.txt see errors and omissions in their coverage across AI developers, as well as contradictions with their terms of service—indicating inefficiencies in the tools used to communicate data intentions.
3. **A divergence in content characteristics between the head and tail of public web-crawled training corpora.** We find the largest web-based sources of public training data have significantly higher rates of user content, multi-modal content, and monetized content, but only slightly less sensitive/explicit content. Top web domains comprise news, encyclopedias, and social media sites, as compared to the many organization websites, blogs, and e-commerce websites in the long tail of web sources.

## 2   Methodology

AI models that are highly performant on tasks in language [18], images [21, 22, 23], video [24, 25, 26], and even audio [27, 28] increasingly depend on massive web-sourced training datasets. These datasets are collected using web crawlers—agents that navigate the web, accessing and retrieving web pages without human intervention. While these robots are essential for a variety of applications, including search engines, studying the internet (i.e., archiving), and link verification tools; recently they have also become the backbone of AI training data collection [29, 30].

In our study, we focus on three popular, open-source, and permissively licensed data sources which are derived from Common Crawl, the largest publicly available crawl of the web, which has collected and stored hundreds of billions of web pages since 2008. For each web-based data source, we sample the web domains from which it was created, and extensively human-annotate their properties. Our analysis examines a snapshot of the present, as well as longitudinal changes across time, to understand how ecosystem norms have evolved.

**Data sources**   The data sources used for our study are C4 [18], RefinedWeb [19], and Dolma [20]. These data sources each have 100k-1M+ downloads, are the primary component in most modern foundation models [30, 1, 31], and are also widely used to derive other popular datasets [32, 33, 34]. Common Crawl is released on a monthly basis, and, as seen in Table 1, each data source is based on a

| ATTRIBUTE | DETAILS | COLLECT |
|---|---|---|
| **Content Modalities** | Whether the web domain has images, videos, and standalone audio in addition to text. | ✎ |
| **User Content** | Whether the web domain hosts primarily content provided by users, such as forums, blog hosting, and social media websites. | ✎ |
| **Sensitive Content** | Whether explicit, illicit, pornographic, or hate speech content is clearly present. | ✎ |
| **Paywall** | Whether the web domain has use limits or any access gating behind a paywall. | ✎ |
| **Advertisements** | Whether the web domain has automatic advertisements embedded into any of its pages. | ✎ |
| **Purpose & Service** | The purpose or service(s) of a website? Options: E-commerce, Social Media/Forum, Encyclopedia, Academic, Government, Organization site, News, or Other. | ✎ |
| *Terms & Restrictions* | | |
| **Robots.txt** | A web domain's robots.txt restrictions on crawler agents. We use Google's crawler rules. | 🤖 ⏱ |
| **Terms & Policies** | The terms, content, copyright, and privacy policy pages found for a web domain. | ✎ ⏱ |
| **Crawling & AI Policy** | Do terms restrict both crawling and AI, restrict crawling, restrict only AI, conditionally restricting crawling/AI, or not apply restrictions? | 🤖 ⏱ |
| **Content Use Policy** | Are there content use restrictions. Options: restricted to personal, academic, or non-commercial use, conditionally restricted, or unrestricted. | 🤖 ⏱ |
| **Non-Compete Policy** | Is content use prohibited for developing competing services? | 🤖 ⏱ |

Table 2: The **list of attributes collected for each web domain**, as sampled from C4, Dolma, and RefinedWeb. 🤖 denotes automatic collection, ✎ human annotation, and ⏱ information collected statically and historically from 2016. Full annotation guidelines are given in Appendix C.2.2.

different set of monthly snapshots. Each of these corpora apply various automatic filtering techniques, including removing duplicative pages, low-quality content, and personally identifying information such as addresses.

**Head sample and random sample**   For each data source, we identified and selected the top 2k web domains ranked by their number of tokens. We refer to the resulting 3.95k union of these web domains as $\text{HEAD}_{\text{All}}$. This sample represents the largest, most actively maintained, and critical domains for AI training. For certain analyses, we consider only the head of C4, which we will refer to as $\text{HEAD}_{\text{C4}}$.

| DATA SOURCE | CRAWL DATES | WEB DOMAINS |
|---|---|---|
| C4 | 4/2019 | 15,928,138 |
| REFINEDWEB | 2008 to 2/2023 | 33,210,738 |
| DOLMA | 5/2020 to 6/2023 | 45,246,789 |
| Intersection | | 10,136,147 |

Table 1: Statistics on audited data sources.

We are also interested in how consent preferences have evolved within a wider sample of internet domains. To capture this, we randomly sampled 10K domains ($\text{RANDOM}_{\text{10k}}$) from the intersection of the three corpora (itself totalling 10,136,147 domains). From the 10k sample, we selected a random subset of 2K for human annotation ($\text{RANDOM}_{\text{2k}}$). $\text{RANDOM}_{\text{10k}}$ was sampled from the intersection of domains listed across all three datasets, which means this subset may skew towards more widely-used or high-quality domains.

**Human annotations**   We trained annotators to manually label the websites for their content modalities (e.g. video, text); website purpose(s) (e.g. news, e-commerce); presence of paywalls and embedded advertisements; the text of the terms of service, if any; and other metadata detailed in Table 2. Annotators received individual instructions, frequent quality calibration, and were compensated well above industry standards at $25-$30 per hour. We collected annotations for the entirety of $\text{HEAD}_{\text{All}}$ as well as from the random sample $\text{RANDOM}_{\text{2k}}$. More details on our annotation process are available in Appendix C, and all annotations will be made publicly available for reproducibility and future research.

**Measuring website administrators' intentions**   A goal of our audit is to measure website administrators' intentions for how their sites can be crawled and their content used—including for training AI models. We used the Wayback Machine[1], a digital archive of 835 billion web pages, to collect historical versions of each website's homepage, its Robots Exclusion Protocol (REP), commonly

---

[1] `https://wayback-api.archive.org/`

referred to as its robots.txt file, and its terms of service page. This was collected at monthly intervals, from January 2016 to April 2024.

The REP, first introduced in 1995 and codified in 2022, has become the default mechanism for website owners to indicate to web crawlers what parts of their website, if any, they consent to have crawled [35]. While it is not legally enforceable, it is respected by all major search engines, as it prevents website servers from getting overloaded by crawlers, it allows websites to signal pages that are undesirable to crawl (for example, calendar sites that could lead to infinite loops), and by respecting it, crawlers disincentivize adversarial tactics designed to impede crawlers. Website creators are able to set one set of instructions for all web crawlers or a different instructions for each web crawler. For instance, Google Search respects instructions which specifies the user-agent string "Googlebot" while Common Crawl looks for the user-agent "CCBot."

In our audit, we record the robots.txt instructions for a range of crawlers, but focus our analysis on five AI developers, Google, OpenAI, Anthropic, Cohere, and Meta, as well as non-profit web archival organizations such as Common Crawl and the Internet Archive, which have seen their data taken for AI training. Collectively, we refer to these as "AI Organizations". We classify robots.txt for each crawler in ascending order of restrictions, from no robots.txt present, to sitemaps which support crawlers without limitations, to basic restrictions on a subset of directories, to full restrictions on any crawling of the website. For each corpus, we measure the percentage of "restricted tokens" as the portion of tokens from web domains that fully restrict one or more of the AI Organizations's crawlers. For Terms of Service analysis, we define restricted tokens to simply mean the portion of unusable tokens due to terms that preclude crawling or AI. See Appendix D.2 for the full list of agents and Appendix D.1 for the robots.txt restriction classification taxonomy.

In addition to robots.txt, we recorded the Terms of Service (ToS) and other content and copyright policies for each website. These documents support more nuanced preferences than the REP, and allow for blanket bans on downstream use cases rather than just specification of what data agents are allowed to collect. We used an automatic annotation pipeline (see Appendix D for details) to categorize ToS agreements according to stance towards use of web crawlers and AI training, content use restrictions, and non-compete clauses, in ascending degrees of restrictiveness.

## 3 Findings

### 3.1 The Rise of Restrictions on Open Web Data

To understand the web sources underlying foundation models, we analyze the longitudinal changes in robots.txt and Terms of Service restrictions between January 2016 and April 2024. In Figure 1 the plots depict the percent of tokens present in each category of restriction over time, for the AI Organizations in $HEAD_{C4}$—the largest, most actively maintained, and critical domains for AI training. The fine-grained longitudinal analysis of robots and Terms of Service trends allows us to estimate this time series into the future. We apply seasonal autoregressive integrated moving average (SARIMA) models to generate forecasts of future trends for both the `head sample` and `random subset`, the details of which can be found in Appendix F along with the coefficients, tests, and limitations.

In Figure 2 we measure the restricted tokens, or how many tokens fall into the most restrictive settings for each of robots.txt and Terms of Service, as a portion of the Full Corpus, or $HEAD_{All}$. The intermittent lack of smoothness for Figures 2c and 2d is mainly due to temporal gaps in the Wayback Machine; however the main trends remain visible. We point to Appendix E for additional details regarding methodology. In all analyses we exclude web domains which could not be retrieved from the Wayback Machine, and all proportions are based on the set of web domains which existed in that time period.

These analyses show a clear and systematic rise in restrictions to crawl and train on data, from across the web. We make no assertion regarding whether the prior omission of a robots.txt or restrictions implies consent to use data. To the degree these restrictions are respected, it also foretells a decline in open data, which may impact more than commercial AI developers, or even AI organizations in general. We break down and discuss the findings of this temporal analysis below.

**Web domains are adopting robots.txt and Terms of Service pages to signal preferences.** Figure 1 (Top & Middle) shows from 2016, the portion of web domains in $HEAD_{C4}$ without a robots.txt and

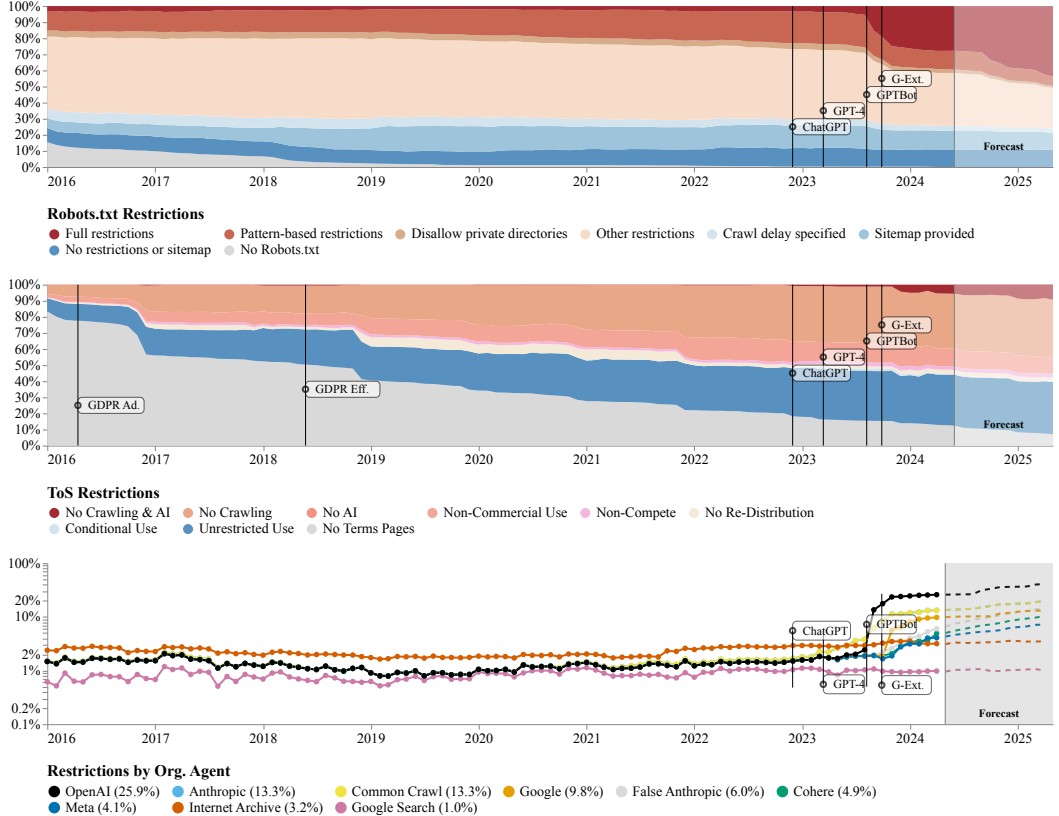

Figure 1: A temporal analysis, from 2016 to April 2024, of the web consent signals in $\text{HEAD}_{C4}$, a sample of the largest and most critical web domains. The colored regions represent the restriction categories as a portion of the total tokens in $\text{HEAD}_{C4}$. We also use SARIMA methods to forecast trends a year into the future. Top: Ascending categories of robots.txt restrictions for the AI Organizations: Google, OpenAI, Anthropic, Cohere, Meta, Common Crawl, and the Internet Archive. Middle: Ascending categories of Terms of Service restrictions (taxonomies described in Table 2). Bottom: A breakdown of robots.txt restrictions by organization, with the April 2024 restriction rates listed in the legend.

Terms of Service has gone from 20% and 80% respectively, to near zero.[2] This reflects an emerging adoption of these practices to signal and protect data intentions.

**Robots.txt crawling restrictions have risen precipitously since mid-2023.** Figure 1(Top) shows the rapid re-distribution of robots.txt restrictions, directly after the introduction of GPTBot and Google-Extended crawler agents. This re-distribution to full restrictions mainly comes from websites with previously moderate restrictions, such as disallowed directories, pattern-based or search page restrictions, and partly from websites with no prior restrictions in their robots.txt.

Across the complete corpora, ~1% of C4, RefinedWeb, and Dolma tokens were restricted in mid 2023, as compared to 5-7% of tokens in April 2024. Among the most critical domains ($\text{HEAD}_{All}$), 20-33% of all tokens are restricted, as compared to <3% one year prior (Figure 2a). From a relative perspective, from Apr 2023 to Apr 2024 these restrictions have risen 500%+ for both C4 and RefinedWeb's full corpus, and 1000%+ for both C4 and RefinedWeb's head sets. Note that these measurements only capture *fully* restricted domains, and the numbers are higher for partially restricted domains.

**AI developers are restricted to widely varying degrees.** Figure 1 (Lower) breaks down the restrictions by AI developers and non-profit organizations. OpenAI crawlers are restricted for 25.9% of tokens in $\text{HEAD}_{C4}$, followed by Anthropic and Common Crawl (13.3%), Google's AI crawler (9.8%),

---

[2]These values may be slightly high, especially for Terms of Service pages, due to gaps in the Wayback Machine.

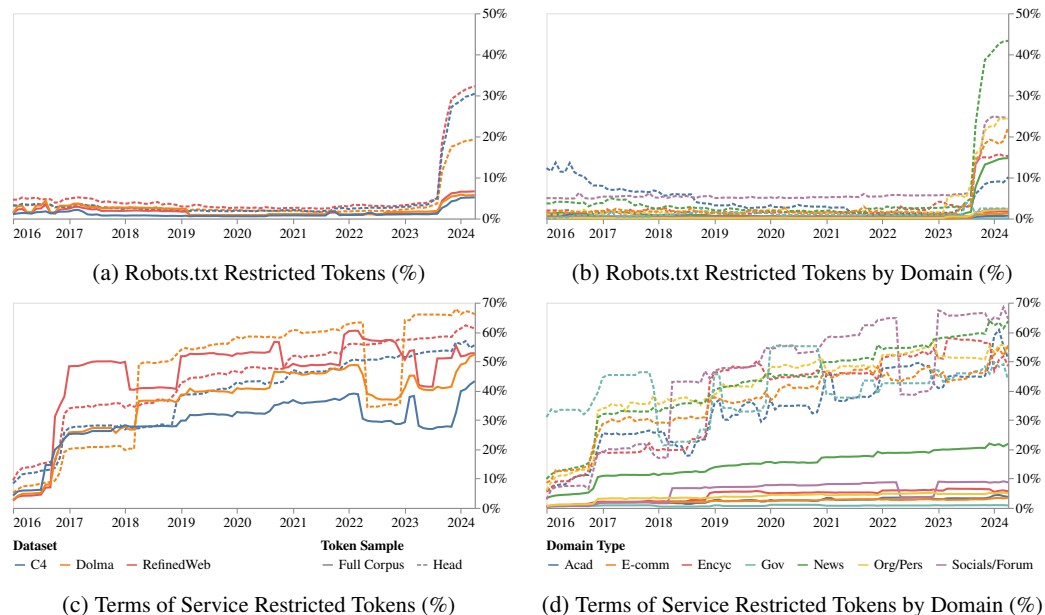

(a) Robots.txt Restricted Tokens (%)

(b) Robots.txt Restricted Tokens by Domain (%)

(c) Terms of Service Restricted Tokens (%)

(d) Terms of Service Restricted Tokens by Domain (%)

Figure 2: A temporal depiction of the percentage of restricted tokens across both the full corpus and the HEAD$_{All}$ sample, which consists of the largest and most critical data sources. The robots.txt analysis (top) and terms of service analysis (bottom) are each broken down by corpus—C4, RefinedWeb, and Dolma (left)—and by domain type, averaged across corpora (right).

and more distantly Cohere (4.9%), Meta (4.1%), the Internet Archive (3.2%), and lastly Google Search's crawler (1.0%). These asymmetries in restrictions have significant differences, and tend to advantage less widely known AI developers. In Subsection 3.2 we discuss these asymmetries and their consequences in more depth.

**Terms of service pages have imposed more anti-crawling and now anti-AI restrictions.** Figure 1 (Middle) illustrates this gradual reformulation of terms pages—with web domains shifting from no terms pages, to those with restrictions on crawling, commercial use, using the data for competing services, or re-distribution. Only in 2024 do we see the wider emergence of terms which specifically mention and restrict the use of their data for generative AI. In the last year, we've seen a 26-53% relative increase in terms-of-service crawling restrictions across C4, RefinedWeb, and Dolma. Figure 2c shows 45-55% of all tokens in these three corpora have a form of data use restriction in their Terms pages. In practice, most automatic crawlers do not heed these terms, though they may provide some avenue for subsequent legal enforcement.[3]

**AI restrictions are driven primarily by news, forums, and social media websites.** For robots.txt, Figure 2b shows nearly 45% of all News website tokens are fully restricted in HEAD$_{All}$, as compared to 3% in 2023. For Terms of Service, Figure 2d shows News website tokens have had a 6% rise in the restricted portion since 2023. Paired with the findings in Table 2, this suggests that the composition of tokens in crawls respecting robots.txt may shift away from news, social media, and forums, and towards organization and e-commerce websites.

**Forecasting trends in the future suggest a continued and significant decline in open and consenting web data sources.** SARIMA forecasts suggest that for just the next year (by April 2025) an additional absolute 2-4% of C4, RefinedWeb, and Dolma tokens will be fully restricted by robots.txt. Equivalently, an additional 7-11% of the highest quality tokens in the head distribution will become restricted. The forecasts for Terms of Service are even starker, with the restricted tokens in the full corpus expected to rise an absolute 6-10% by April 2025. These trends illustrate a systematic rise in restrictions on data sources, which, where enforced or respected, will severely hamper data

---

[3]For instance, see Bogard v. TikTok Inc., No. 3:23-cv-00012-RLY_MJD, 2024 WL 1588423, at *4 (S.D. Ind. Mar. 24, 2024).

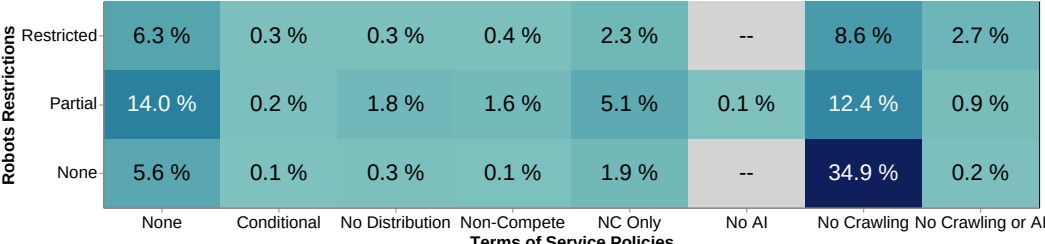

Figure 3: A cross-tabulation of the terms of service policies and robots.txt restrictions for $\text{HEAD}_{\text{C4}}$, measured in percentage of tokens. **We find that these two ways of expressing restrictions on data use for AI** *frequently disagree*, in both what they express and what they can express.

scaling practices in the coming years—practices which have thus far been responsible for remarkable capability improvements.

## 3.2 Inconsistent and Ineffective Communication on AI Consent

In many cases, data providers and rightsholders fail to effectively communicate their preferences on how their data is used by AI systems. We observe robots.txt instructions which allow some AI organizations to crawl while restricting others, references to non-existent crawlers, and contradictions between the robots.txt and Terms of Service. Together, these issues point to the need for better preference signaling protocols.

**Some AI crawlers are allowed, while others are not.** We find not all AI agents are disallowed equally. In Table 3 we estimate the conditional probabilities of each organization's crawler being restricted, conditioned on whether any other AI organization is restricted. Whereas OpenAI and Common Crawl agents are frequently disallowed (in 91.5% and 83.4% of cases where *any* of the organizations are disallowed), the agents of other AI companies, such as Google, Cohere, and Meta are often omitted from robots.txt. The omissions of Cohere, Meta, and other small AI organizations are likely because website administrators are unaware or unable to update their robots.txt to reflect the full list of AI developers. On the other hand, the particularly high omission rates of Internet Archive and Google Search suggest web administrators may be open to more traditional crawler uses like archiving and search engines, even as they seek to restrict AI usage. A full confusion matrix showing the correlation between restrictions for each user agent is provided in Appendix Figure 5.

| ORGANIZATION | REST. (%) |
|---|---|
| OPENAI | 91.5 |
| COMMON CRAWL | 83.4 |
| ANTHROPIC | 83.4 |
| GOOGLE EXTENDED | 72.0 |
| FALSE ANTHROPIC | 61.6 |
| COHERE | 52.3 |
| META | 52.2 |
| INTERNET ARCHIVE | 32.3 |
| GOOGLE SEARCH | 17.1 |

Table 3: The % each org's crawler agents are **restricted** if at least one other org in this pool is restricted. Gray indicates crawlers with a primary purpose other than AI training data.

**Unrecognized crawler agents cause incorrect specifications.** We find several instances where robots.txt refers to user agents that the companies do not recognize. For instance, 4.5% of websites disallowed the unrecognized user agents ANTHROPIC-AI or CLAUDE-WEB (documented as FALSE ANTHROPIC), but not the documented agent for Anthropic's crawler, CLAUDEBOT. The origin and reason for these unrecognized user agents remains unclear—Anthropic reports no use of them. These inconsistencies and omissions across AI agents suggest that a significant burden is placed on the domain creator to understand evolving agent specifications across (a growing number of) developers. AI crawler standardization could address these challenges in consent/preference signaling.

**Contradictions exist between robots.txt and ToS.** The Robots Exclusion Protocol (REP) is a guideline for web crawlers, while a website's terms of service is a legal agreement between the website and users of the site. The benefit of the REP is its machine-readability. However, its rigid structure, created in 1995, limits what signals it can convey. In contrast, a ToS can communicate

| Variable | URL Group | | | | Stats | Pct. Tokens in Corpus | | |
|---|---|---|---|---|---|---|---|---|
| | Top 100 | Top 500 | Top 2000 | Random | Diff | C4 | RW | Dolma |
| Restrictive Robots.txt | **38.4** | **35.0** | **26.5** | 3.4 | +23.1 | 5.0±1.5 | 6.6±2.3 | 5.6±1.9 |
| Restrictive Terms | **64.1** | **61.0** | **51.2** | 15.7 | +35.5 | 43.2±15.2 | 52.8±30.3 | 52.3±15.4 |
| User Content | **21.3** | 19.1 | **19.4** | 15.1 | +4.4 | 27.9±12.3 | 39.8±32.8 | 37.3±16.7 |
| Paywall | **31.8** | **31.3** | **24.6** | 1.6 | +23.0 | 4.1±1.1 | 4.9±0.4 | 10.8±1.2 |
| Ads | **54.6** | **61.4** | **53.2** | 5.4 | +47.9 | 23.5±12.6 | 44.8±34.4 | 34.8±18.1 |
| Modality: Image | 96.8 | 97.0 | 96.7 | 95.0 | +1.7 | 97.7±2.3 | 98.6±0.9 | 97.5±1.9 |
| Modality: Video | **87.0** | **78.8** | **58.7** | 18.9 | +39.8 | 32.9±14.2 | 27.0±14.7 | 35.4±10.6 |
| Modality: Audio | **80.7** | **68.3** | **41.8** | 3.4 | +38.4 | 21.2±14.7 | 12.5±6.3 | 20.5±6.7 |
| Sensitive Content | 0.0 | 0.4 | 1.1 | 0.6 | +0.5 | 0.8±1.0 | 0.2±0.4 | 1.8±3.0 |
| *Web Domain Service & Purpose* | | | | | | | | |
| Academic | **14.1** | **10.1** | **9.8** | 3.8 | +6.0 | 3.1±1.6 | 2.6±1.2 | 3.0±0.7 |
| Blogs | **2.6** | **2.9** | **3.9** | 15.1 | -11.2 | 23.2±11.3 | 16.3±16.0 | 20.1±11.9 |
| E-Commerce | 8.4 | 9.9 | 10.1 | 10.6 | -0.5 | 20.0±17.8 | 32.6±37.6 | 17.7±19.1 |
| Encyclopedia/Database | **20.5** | **13.2** | **11.1** | 0.4 | +10.7 | 3.5±3.4 | 5.8±9.8 | 5.1±5.8 |
| Government | **3.2** | **2.8** | **2.8** | 1.1 | +1.7 | 0.9±0.9 | 0.9±0.8 | 0.8±0.6 |
| News/Periodicals | **45.6** | **53.3** | **50.0** | 5.3 | +44.7 | 11.5±3.9 | 16.8±10.8 | 22.9±10.9 |
| Org/Personal Website | **15.3** | **13.2** | **12.7** | 71.2 | -58.5 | 48.5±13.3 | 57.3±24.2 | 46.3±14.2 |
| Social Media/Forums | **9.4** | **9.3** | **11.8** | 1.6 | +10.1 | 5.1±4.8 | 5.4±8.9 | 14.9±8.3 |
| Other | **15.0** | **10.9** | **11.8** | 4.3 | +7.4 | 4.7±2.7 | 2.8±1.3 | 3.7±2.0 |

Table 4: **Mean incidence rates of web source features across C4, RefinedWeb, and Dolma.** We measure incidence rates for the top 100, 500, and 2000 URLs, ranked by number of tokens, as well as the random sample. The 'Diff' column reports the % difference between the top 2k and random samples. We test for significant differences between the overall corpus and each of the top-100, top-500 and top-2000 sets with a Bonferroni-corrected two-sided permutation test, where differences significant at the Bonferroni-corrected $5\sigma$ level are indicated in bold. We also estimate the percentage of tokens in each corpus, C4, RefinedWeb, and Dolma, for which the web feature is present ($\pm$ 95% bootstrap CI shown in gray), by computing the final percentage of tokens based on the estimate for the unobserved population (from the random sample), and the observed head sample.

rich and nuanced policies in natural language. Without a robots.txt, a ToS lacks practical deterrence of unwanted crawling. Inversely, without a ToS, a robots.txt may lack any plausible means of enforcement [36]. We found that in many cases, websites' robots.txt implementations fail to capture the intentions specified in their terms of service.

In Figure 3, we illustrate the distribution of terms and REP use criteria (the taxonomy is defined in Table 2 and broken down in detail in Appendix D). Common use criteria expressed in modern ToS pages include prohibitions specifically on commercial use, conditional use limiting actions such as third-party re-posting, non-compete criteria, or specific prohibitions only against "AI", but not against crawling for search engines. We also see many websites write anti-crawling terms but have no robots.txt file (35.1%), or have no ToS but a restrictive robots.txt (20.3%) that disallows at least some crawlers. Terms specifying only non-commercial uses are also often paired with fully or partially restrictive robots.txt files, which may unintentionally limit academic web crawlers, as a side effect of deterring corporate use. Another formidable challenge is that websites currently have to list every search engine or AI user agent they want to restrict. Empirical evidence from both Figure 5 and Figure 3 suggests the absence of REP expressivity and standardization for AI is leading to inconsistent or unintended signals that fail to reflect intended preferences.

### 3.3 Correlating Features of Web Data

What does web data actually look like? Prior work has measured the characteristics of web-derived datasets, for the presence of artifacts [8, 11], undesirable text and images [37, 12], demographic biases [9], and quality discrepancies across languages [38]. We expand upon these analyses by measuring what web data sources look like *before* they have been neatly processed into AI training datasets. We measure the presence of multi-modal content, user-derived content, website monetization schemes, and sensitive content on the most well-represented web domains on the internet (HEAD_All) and on a

random sample of domains (RANDOM$_{2k}$). We also annotate the services provided and purpose of each web domain.

**Most of the web is composed of organizational/personal websites, and blogs, however the head distribution is disproportionately news, forums, and encyclopedias.** Table 4 shows several notable and statistically significant differences between head distribution (HEAD$_{All}$) and tail distribution (RANDOM$_{2k}$) of web domains. HEAD$_{All}$ comprises mostly news, social media/forums, and encyclopedias (72.9%), in contrast to the long tail data in RANDOM$_{2k}$, which is dominated by personal or organization websites, blogs and E-commerce sites (97%). Academic and government content is also proportionately more common in the head distribution. Note however that though they are all derived from Common Crawl snapshots, C4, RefinedWeb, and Dolma all show variations in their source compositions—highlighting the importance of curation choices.

**The head distribution of domains is more multimodal, and heavily monetized.** We observe that HEAD$_{All}$ web domains are much more heavily monetized through ads (+47.5%) and paywalls (+24.1%). Accordingly, they also have significantly greater restrictions from both robots.txt (+22.5%) and terms of service (+35.3%). This greater prevalence of monetization and restrictions likely corresponds to the higher quality and heterogeneity of content usually produced by news, periodicals, forums, and databases, which are more common in HEAD$_{All}$. This is reflected by the higher proportions of image (+4.4%), video (+39.8%), and audio content (+38.4%) than the rest of the web. Interestingly, the fraction of user-generated content and sensitive content between the head and tail distributions is less pronounced. Crawlers which respect the restrictions that occur far more frequently in HEAD$_{All}$ will increasingly lose access to the most multimodal, highly curated, and up-to-date content sources.

## 4 Discussion

**Contextual Background and Motivation**    Website content permissions are rapidly changing within a broader context. Dataset creators and model developers have conflicting interests, particularly financial ones, as creators seek credit and compensation for their work [39]. AI systems' potential to replace human labor [40] leads many creators to oppose using their data in potentially competing systems [4]. On top of these considerations, copyright law remains unclear on AI and training data issues, including generated text authorship (because a model is not a legal person), fair use boundaries, and infringement thresholds [41]. Legal clarification will require extensive work from regulatory agencies, courts and legislatures. All parties lack the certainty and protection that well established law provides. This uncertainty is amplified by the abandonment of pre-AI conventions for scraped data use [42, 43] in current AI applications [6, 44]. These legal and economic uncertainties drive creators toward increased data restrictions.

**The web-sourced AI data commons is rapidly becoming more restricted.**    The web has acted as the primary "data commons" for general-purpose AI. Its scale and heterogeneity have become fundamental to advances in capabilities. However, our results show web domains are rapidly restricting crawling and use of their content for AI. In less than a year, ~5% of the tokens in C4 and other major corpora have recently become restricted by robots.txt. And nearly 45% of these tokens now carry some form of restrictions from the domain's terms of service. If these rising restrictions are respected by model developers (as many developers claim) or are legally enforced, the availability of high-quality pretraining sources will rapidly diminish.

**The rise in restrictions will skew data representativeness, freshness, and scaling laws.**    Prior work has emphasized scaling data as essential to improving frontier model capabilities [45, 46]. While the trend toward increasingly restricted data will respect content creators' intentions, it will also challenge these data scaling laws [45, 46]. Not only do these restrictions reduce the scale of available data, they also change the composition (away from news and forums), diversity, and representativeness of training data—biasing this data toward older and less fresh content.

Recently, multiple AI developers have been accused of bypassing robots.txt opt-outs to scrape publisher websites [47, 48]. While it is not possible to confirm, in each case it appears AI systems may be distinguishing between crawling data for training, and crawling data to retrieve information for user questions at inference time. One of the few, OpenAI has two crawler agents, GPTBot for training, and ChatGPT-User for live browsing plugins (see Table 5). Other companies may simply not

be registering their inference time crawlers for opt-outs. This circumvention may allow developers to directly attribute the retrieved web pages, as well as better achieve data representativeness, freshness, and approximate the scaling laws had they trained on it. However, creators may feel this violates the spirit of the opt-outs, especially if the opportunity to attribute sources is not taken.

**The web needs better protocols to express intentions and consent.** The REP places an immense burden on website owners to correctly anticipate all agents who may crawl their domain for undesired downstream use cases. We consistently find this leads to protocol implementations that don't reflect intended preferences. An alternative scheme might give website owners control over *how* their webpages are used rather than *who* can use them. This would involve standardizing a taxonomy that better represents downstream use cases, e.g. allowing domain owners to specify that web crawling only be used for search engines, or only for non-commercial AI, or only for AI that attributes outputs to their source data. New commands could also set extended restriction periods, because dynamic sites may want to block crawlers for extended periods of time (e.g. for news organizations to protect their most recent work). Ultimately, a new protocol should lead to website owners having greater capacity to self-sort consensual from non-consensual uses, implementing machine-readable instructions that approximate the natural language instructions in their terms of service.

**Rising expressions of non-consent will affect non-profits, archives, and academic researchers.** A new wave of robots.txt and terms of service pages have notdistinguish (or cannot distinguish) the various uses of their data. For instance, having to individually prohibit a plethora of AI crawlers has motivated many domains to switch to a blanket prohibition of any crawling with the wildcard "*" marker. Domains have also limited crawlers from non-profit archives such as the Common Crawl Foundation or Internet Archive, in order to prevent other organizations from downloaded their data for training. However, these archives are also used for non-commercial uses of AI, as well as academic research, knowledge, and accountability, well beyond the scope of AI. For instance, the Common Crawl is reported to be cited in 10,000+ research articles from varying fields.[4] This tension between data creators and, predominantly, commercial AI developers has left academic and non-commercial interests as secondary victims. As web consent continues to evolve, we believe it is essential that these widely used facilities not be marginalized or severely hampered.

## 5   Related Work

Prior work has conducted large scale audits of the provenance, quality, biases, and characteristics of AI training data, for pretraining text [9, 10, 8, 49], finetuning text [11], as well as multimodal datasets [12, 50, 51, 52, 53], and challenges in data development [54]. Recent work has looked at collecting non-copyrighted data [13], interpreting the legal implications of fair use for AI data [55, 56], and forecasting future data constraints [46]. However, there is little work inspecting the evolution of consent signals on AI data. Prior research has attempted to understand link decay on the web [57], Common Crawl's collection process [58], and web crawlers' evolving behavior and implications [59, 60, 61, 62, 63]. Initial news reports have begun to investigate the rates of blocking AI web crawlers for general websites [64] and news publishers [65], setting the stage for our more rigorous analysis. The dearth of AI dataset documentation [66, 67, 68, 69] has been highlighted as a challenge for understanding model behavior [70, 71, 72, 73, 74], reproducibility, consent, and authenticity [75].

## 6   Conclusion

In this work, we presented the first large-scale audit of the web sources underlying the massive training corpora for modern, general-purpose AI. Our audit of $14,000$ web domains provides a view the changing nature of crawlable content, consent norms, and points to daunting trends for the future openness of the highest quality data used to train AI. The inconsistencies and omissions between robots.txt and terms of service pages suggest a data ecosystem ill-equipped to signal or enforce preferences. Lastly, we uncover distributional mismatches in the documented real uses of AI systems and their underlying data. We release all our collected annotations and analysis, with the hope that future work will further investigate the provenance, consent, and composition of these fundamental ingredients in AI systems.[5]

---

[4] https://commoncrawl.org/

[5] https://github.com/Data-Provenance-Initiative/Data-Provenance-Collection

## Acknowledgments and Disclosure of Funding

This research was conducted by the Data Provenance Initiative, a collective of independent and academic researchers volunteering their time to data transparency projects. The Data Provenance Initiative is supported by the Mozilla Data Futures Lab Infrastructure Fund.

We would like to thank Arvind Narayanan, Stefan Baack, Aviya Skowron, Cullen Miller, Greg Lindahl, Pedro Ortiz Suarez, and Anna Tumadóttir for their insightful feedback and guidance.

We are grateful to Donna Sousa, Dabashis Kundu Shento, Umm E. Habiba, and Fazia Khan, as well as several other annotators who wish to remain anonymous, on Upwork for their manual annotations.

Lastly, we would also like to thank the ML Collective community for the generous computational support on experiments.

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

## Contributions

Here we break down contributions to this work. Contributors are listed alphabetically, except for team leads who are placed first.

- **Annotation Process Design for Web Domain Services**  Shayne Longpre (lead), Robert Mahari (lead), Hanlin Li (lead),Ahmad Mustafa Anis, Deividas Mataciunas, Diganta Misra, Emad Alghamdi, Enrico Shippole, Hamidah Oderinwale, Jianguo Zhang, Joanna Materzynska, Kevin Klyman, Kun Qian, Kush Tiwary, Lester Miranda, Manan Dey, Manuel Cherep, Minnie Liang, Mohammed Hamdy, Nayan Saxena, Nikhil Singh, Niklas Muennighoff, Naana Obeng-Marnu, Robert Mahari, Seonghyeon Ye, Seungone Kim, Shayne Longpre, Shrestha Mohanty, Tobin South, Vipul Gupta, Vivek Sharma, Vu Minh Chien, William Brannon, Xuhui Zhou, Yizhi Li, An Dinh, Ariel Lee, Campbell Lund, Caroline Chitongo, Christopher Klamm, Cole Hunter, Da Yin, Damien Sileo, Hailey Schoelkopf

- **Annotation Process Design for Web Domain Characteristics**  Shayne Longpre (lead), Robert Mahari (lead)

- **Annotation Process Design for Terms of Service**  Robert Mahari (lead); Hamidah Oderinwale (lead), Campbell Lund (lead), Shayne Longpre

- **Annotations & Annotation Quality Review**  Robert Mahari (lead), Shayne Longpre (lead), Jad Kabbara (lead), Ahmad Mustafa Anis, William Brannon, Caroline Chitongo, Vu Minh Chien, Manan Dey, An Dinh, Da Yin, Vipul Gupta, Mohammed Hamdy, Cole Hunter, Daphne Ippolito, Jad Kabbara, Christopher Klamm, Kevin Klyman, Ariel Lee, Minnie Liang, Hanlin Li, Lester Miranda, Shrestha Mohanty, Niklas Muennighoff, Seungone Kim, Damien Sileo, Hailey Schoelkopf, Enrico Shippole, Tobin South, Nayan Saxena, Xuhui Zhou

- **Data Corpus Collection**  Tobin South (lead)

- **Wayback Machine Data Collection**  Ariel Lee (lead)

- **Robots.txt Longitudinal Analysis**  Ariel Lee (lead), Shayne Longpre (lead), Nikhil Singh (lead), Nayan Saxena, Tobin South,

- **Terms of Service Longitudinal Analysis**  Ariel Lee (lead), Shayne Longpre (lead)

- **Trend Forecasting**  Ariel Lee (lead)

- **Robots.txt and ToS Comparisons**  Shayne Longpre (lead), William Brannon (lead), Campbell Lund, Ariel Lee

- **Web Domain Characteristics Analysis**  William Brannon (lead), Shayne Longpre (lead)

- **Annotation Process Design for WildChat**  Shayne Longpre (lead), Nayan Saxena (lead)

- **WildChat vs Web Domain Analysis**  Shayne Longpre (lead), Manuel Cherep (lead), Campbell Lund, Ariel Lee, Nayan Saxena

- **Writing**  Shayne Longpre (lead), Jad Kabbara (lead), Robert Mahari (lead), Daphne Ippolito (lead), Sara Hooker (lead)

- **Legal Analysis**  Robert Mahari (lead), Luis Villa

- **Visualizations & Visual Data Analysis**  Naana Obeng-Marnu (lead), Nikhil Singh (lead), Shayne Longpre (lead), William Brannon (lead)

- **Senior Advisors**  Stella Biderman, Daphne Ippolito, Sara Hooker, Jad Kabbara, Hanlin Li, Sandy Pentland, Luis Villa, Caiming Xiong

# Appendix

## Table of Contents

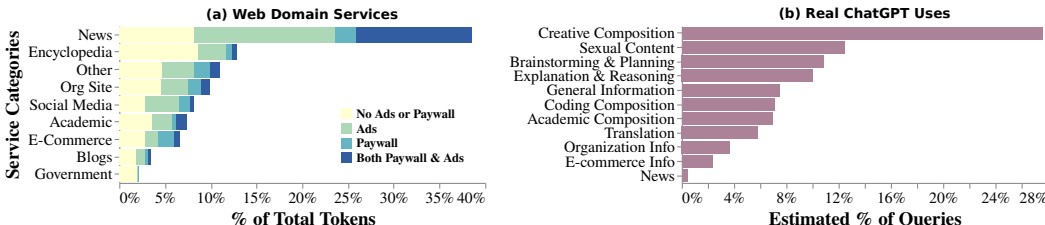

Figure 4: **The most common services provided by web domains in HEAD$_{C4}$ do not match real ChatGPT use cases from WildChat user logs.** Left: We measure the proportion of tokens in HEAD$_{C4}$ dedicated to each type of web service, and the degree to which they are monetized via paywalls and ads. Right: We measure the proportion of each type of user query in WildChat.

## A    Misalignment Between Real-world AI Usage and Web Data

### A.1    WildChat Findings

In this section, we measure the degree of alignment between real world uses of ChatGPT and the content in the webcrawls that form the bulk of AI training. For each web domain in HEAD$_{All}$, we had annotators label the services provided by the website, as well as the presence of some monetization, such as a paywall or automatic ads. We compare these services against the services that real-world users solicit in their interactions with conversational AI systems. We use WildChat, a recent set of 1 million user conversations with ChatGPT [76], collected through a HuggingFace Space wrapper around OpenAI services. We randomly sampled 100 conversation logs from WildChat, which the paper authors manually clustered by the type of tasks or goals conveyed by each conversation, with the goal of relating the core function of these conversations with the services provided by the websites crawled in training. Subsequently, we used GPT-4o to label 1k randomly selected conversations from the WildChat dataset; these conversations were labelled using the taxonomy we developed to categorize websites. Further details on the taxonomy and labelling procedure can be found in Appendix D.6.

**Apparent uses of ChatGPT are misaligned with the popular web domains language models are trained on.** Figure 4(a) shows the distribution of services provided by the web domains, broken down by whether those domains are monetized. In contrast, Figure 4(b) shows how ChatGPT is used in the real world. The way that users interact with ChatGPT is different in important ways from the types of content that is most frequently represented in publicly available web-based training datasets. For instance, in over 30% of conversations, users request creative compositions such as fictional story writing or continuation, role-playing, or poetry. However, creative writing is poorly represented among the web data used for model training. These results may provide evidence for where models trained exclusively on unstructured internet data are most "unaligned" with how real users want to use generative AI [77]. Language models trained only on web data are known to struggle to understand the structure of discourse and underperform models trained with instruction finetuning and preference training on highly curated data [78, 79, 80]. The misalignment between real use cases and web crawled data may suggest the key areas of model distributional misalignment, as well as inform future data collection efforts based on real-world uses.

**Sexual role-play appears to be a prevalent use of ChatGPT, despite being mostly removed from common public datasets.** Whereas sensitive (e.g. sexual) content represents $< 1\%$ of the web domains in HEAD$_{C4}$ (see Table 4), sexual role-play represents 12% of all recorded user interactions in WildChat. All the public datasets we consider—C4, RefinedWeb, and Dolma—have undergone some form of filtering to remove illegal or sexually explicit content, as training on such content introduces potential liability concerns; the web, in general, is known to have high portions of sexually explicit content [12, 81]. OpenAI states in the GPT-4 technical report that it also filtered its training data for harmful content [82]. In addition to filtering web-derived training data, OpenAI's models are further trained to refuse requests that violate OpenAI's Usage Policies.[6] OpenAI's Usage Policies prohibit "sexually explicit or suggestive content" with respect to minors, or re-distribution that may harm others; however, there is ambiguity as to whether this would cover all user requests for sexual role-

---

[6] https://openai.com/policies/usage-policies/

play [83]. For instance, the GPT-4 technical report makes a distinction in model refusal instructions between erotic and non-erotic sexual content, "(e.g. literary or artistic value) and contextualized sexual content (e.g. medical)" [82].

Sexual-related uses of AI are a topic of ongoing debate within the scientific community [84, 85, 86], and rules differ by company, service, and jurisdiction. In a review of 30 generative AI developers' acceptable use policies, Klyman [83] finds that OpenAI's policies are not among the most restrictive with respect to sexual content; while OpenAI has a blanket ban on "sexually explicit or suggestive content," other companies' acceptable use policies also explicitly prohibit "erotic content," "adult content," "pornography," "nudity," and "sexual fetishes" [87, 88, 89]. However, harsher restrictions on sexual content come with tradeoffs, as more heavily safety-tuned language models may then be less able to direct users to resources about sex education or generate fictional stories with PG-13 type content.

**Common ChatGPT uses appear distinct from the uses of commercialized web sources.** Figure 4 shows that a significant portion of tokens in $\text{HEAD}_{C4}$ are from web domains with ads, paywalls, or both—in other words they are the most commercialized. However, while news websites (the mostly highly commercialized category) comprise nearly 40% of all tokens in $\text{HEAD}_{C4}$, fewer than 1% of ChatGPT queries appear to be related to news or current affairs. It also shows that news websites have the highest instance of ads, paywalls, or both—in other words, they are the most commercialized. Our observations suggest that real-world use cases of ChatGPT are not necessarily directly related to the most prevalent, commercialized content on the web. This finding has interesting implications for the use of AI in industries with web-based services, such as journalism, or for US copyright analysis, which evaluates how the secondary use of a protected work (training AI models) affects the potential market for the original use of the work (see 17 U.S.C §107).

## A.2    Implications for Copyright and Fair Use

Analysis of copyright infringement, including fair use, includes a four factor analysis. One analysis evaluates how the use of a protected work (e.g., to train AI models) affects the potential market for the original work (see 17 U.S.C §107). To investigate this question broadly, we document the major use settings of primary training domains, and compare those to the real use cases found in WildChat ( Appendix A.1). We find that while News domains dominate as a source of data, ChatGPT is not currently used often for news—instead uses like creative compositions (such as role-play or fiction writing), sexual role-play, brainstorming, or general information requests are most common. While there exist several limitations to this analysis, outlined in Appendix A.1, the mismatch in use cases between training data and popular chatbots might suggest that AI chatbots are not directly competing with many of their training sources. We caution against over-interpreting these results to suggest a stronger case for fair use, as we believe future work is necessary to substantiate these findings and their relation to nuanced legal discussions.

## A.3    Economic and Privacy Concerns Affect Data Creation and Protection

The content on the internet was not created to be used for training AI models. Its use for this purpose is already resulting in changing incentives around content creation, especially in cases where generative AI competes with the original sources of content. As we show in Figure 4, large portions of today's internet are owned by commercial interests, with sites that are locked behind paywalls or financed by advertisements. We expect small-scale content providers, who are less resourced to protect themselves from undesired crawling, may opt out of the web entirely, or move to posting on walled, content websites. Further, some of these small content providers may not be aware that their web-data is at risk of crawling, or that protocols to protect their data against unwanted uses even exist. In this case, groups of creators with the least knowledge of the internet are at the highest risk of data rights and privacy breaches.

If we don't develop better mechanisms to give website owners control over how their data is used, we should expect to see further decreases in the open web. This means more websites locking their data behind login or paywalls to prevent it being trained on, or risking the rights to privacy and consent of groups without technical knowledge of web consent protocols.

## A.4 Limitations

We believe these observations provide empirical evidence for the (mis)alignment between AI uses and web-derived training data. However, our observations come with significant caveats. The WildChat [76] dataset may not include a representative sample of how people interact with language models. Not only does it include conversations only with a specific model (ChatGPT), but the WildChat proxy service is hosted on a technical website, HuggingFace Spaces, which could suggest a more technical user base, or one more likely to audit ChatGPT for inappropriate uses. Model uses change both by time and product; our analysis is specific to the model interactions collected in WildChat between April 9, 2023, at 12 AM to May 1, 2024, using the GPT3.5-Turbo and GPT-4 APIs. Different AI products are likely to have different use distributions, and usage patterns will inevitably change over time. Finally, the use taxonomy, both for web domains and WildChat uses, were developed based on a manual, iterative process that is limited in its granularity. Lastly, it is possible that data/information from News web domains could be used in responses for non-News classifications in WildChat, e.g. General Information. This would be exceedingly difficult to measure, and merits analysis in future work.

# B  Impact & Ethics Statement

It is important to note that these trends are relevant to massive, data-hungry models in their current state and usage. A shift to more learning-efficient models, proprietary datasets, and/or smaller models could promise some solutions to invasive data extraction practices.

Consent to copy, use and train on data is a complex issue. First, the robots.txt and Terms of Service that communicate these intentions are owned by the web administrators, which are often imperfect proxies for the actual copyright holders. For instance, social media websites or forums often host content that was originally created or belongs to others. This is pervasive across the web. And there are insufficient tools to attribute all content to their copyright holders, or disentangle consenting from non-consenting use content—indeed that is partly demonstrated by this work. As such, it is important to recognize that robots.txt and Terms of Service have become the status quo out of practicality, though they suffer from limitations in ownership, and effective communication of intentions.

Additionally, while many data preference signals exist, which ones should be enforceable and how they should be enforced both remain open questions, legally and ethically. Data crawling restrictions can be motivated by intentions to protect copyright holders, privacy, or a desire to monetize the data themselves. Some of these motivations may not override the competing right for humans to collect public web material, for study, or non-commercial purposes. And, some have argued that humans, and by extension machines, have the "right to read and learn" from open web data [90]. The laws, ethics, and best practices that emerge around these conflicting goals will impact the future efficacy of AI technologies, the types of organizations that are able to acquire sufficient data to compete in frontier model development, as well as the economy of creators from which these datasets are sourced. In this work, we do not prescribe legal or ethical answers, but describe the precise and evolving nature of preference signals on the web. While we advocate for more protocols and mechanisms that enable more effective communication of these intentions, we leave the adherence to these intentions as a broader question for readers, developers, and legislators.

Our findings suggest websites and data creators are rapidly working to secure their data against crawling practices which often do not respect creator consent or provide attribution. Further, the use of this data for generative AI could impact the creators' livelihoods, especially in news or the arts, as discussed in Section 4. While the increase in websites expressing their data-use preferences is positive, our analysis exposes several on-going challenges. First, the burden the machine-readable robots.txt standard places on these sites to enumerate all AI crawlers (without blocking other crawlers like search engines) is onerous—leaving them with significant gaps in their intended coverage. Second, our results suggest that this existing standard is insufficient for websites to express their preferences as accurately as in their terms of service. The inability to specify permissions by type of use (non-commercial, attribution, etc), rather than by individual crawler, exposes less invasive and usually more preference-respecting data uses like academic research to restrictions that may not even be intended for them. On the other hand, the lack of legal enforcement of robots.txt suggests restrictions intended for corporations may not be heeded anyway. Third, websites are not always the copyright holders, meaning existing standards may empower web-platforms at the expense of

creators. Fourth, it remains ambiguous whether new restrictions can apply retroactively. Altogether, the expression of website and creator preferences is a positive trend, but it still lacks certainty around enforcement or future practices.

Further, indiscriminate data scraping, without consent from websites or copyright holders, may not only deny credit and compensation to data creators, but also poses broader data rights and privacy concerns. Sensitive and private information, even about non-users of a website who did not share their own information with it, may become more widely available, or be cross-referenced or exposed in unanticipated situations. And there are only limited mechanisms for users to ask that their information be removed or unlearned from large models, after expensive training runs have completed. These practices may strongly alter the culture around consent, privacy, and sharing of information on the web more broadly. Data creators may undertake more rigorous and difficult measures to protect their data, such as anti-crawling or data-poisoning methods, or work to eliminate any online presence of their data at all. Lastly, it is worth highlighting that large-scale data collection and model pretraining have significant environmental costs, including water and energy usage.

## C  Human Annotation Methodology Details

### C.1  Details on Crowdworkers

Many of the annotations we rely on were provided by a group of crowdworkers. We engaged in an extensive and iterative training process to ensure that each worker was comfortable with the task and to guarantee consistency across them. We employed a total of 14 crowd source workers from six countries: Pakistan (8), Bangladesh (2), Philippines (1), and USA (1). We paid a total of $6,972 to annotate 14,228 rows of data, with a mean of $498 per worker. Annotator pay by country is as follows, with the average income in parentheses following our hourly rate: Pakistan - $24.05/hour ($3.13/hour); Bangladesh - $35.92/hour ($3.85/hour); USA - $25.25/hour ($38.98/hour); and Philippines - $25.05/hour ($5.37/hour). Our data annotation process involved daily check-ins, review of every 100-200 annotations, and feedback to ensure quality and consistency.

### C.2  Human Annotation Guidelines

This section lays out the annotation guidelines used for our pretraining data collection, both for annotations carried out by authors (in Appendix C.2.1) and for those carried out by crowdworkers (in Appendix C.2.2).

#### C.2.1  Web Source Annotations (Authors)

Some websites, that were crawled in earlier years, have since been shutdown and no longer work. We record this and exclude them from our analysis.

> **Instructions for Website Issue**
>
> Some websites have been sold or shut down since the scrape. In these cases, check the box for website issues and don't continue.

For **User Content**, we aimed to differentiate websites with significant portions of unmoderated user content from those that are primarily comprised of content curated by the website administrators. Over the course of annotating we found that the "Yes (strong moderation)" annotation label was often used for news and encyclopedic sources which did accept some (usually moderated) user content, but were most similar to websites without any user content ("No" label). In contrast, the "Yes (weak moderation)" websites tended to include websites with significant degrees of raw user-generated content, such as from social media websites, forums, or review sites. As such, in the paper we group "No" and "Yes (strong moderation)" as not accepting significant user content, whereas "Yes (weak moderation)" does.

**Instructions for User Content**

Is there a non-negligible amount of content on the website that comes from third-party users, instead of the website host? Options:

**Yes (strong moderation)** – there is content from third-parties, but it is strongly moderated/curated, either by the host, or by a review system. E.g. Wikipedia, academic journal websites, or NYTimes, since it has a comments section, but it is carefully moderated.

**Yes (weak moderation)** – there is content from third-parties, that is only weakly moderated. E.g. reddit, stackoverflow, youtube, ecommerce comment/review websites, or very low-quality news sites that have unrefined op-eds and comments sections that appear completely unmoderated.

**No** – all (or the vast majority) of website content is provided or well curated by the host. E.g. company websites, patent records, government databases.

**Instructions for "Website Description"**

Write a short phrase that describes the purpose and domain of the website. The goal is to help us cluster and categorize websites by their content domain (the type(s) of content/topics they contain e.g. legal, biomedical, books) as well as the type/purpose of service the website is providing (e.g. news, social media, exams, ecommerce, etc). While there is some overlap, the first helps to distinguish where the training data might be useful, whereas the second determines the purpose of the website, for copyright infringement questions.
Make sure the short phrase captures all major elements of a website's purpose and content, as there can be multiple, and is as precise as possible. Here are some examples:

- "Lifestyle blog about travel"
- "E-commerce for appliances and product reviews"
- "Video game news, forums, art, and retail"
- "Government database of parliamentary recordings and legislative documents"
- "Informal blog site for baking recipes"

The content domain and type of service categories should be easily inferred from the website description.

The purpose of the **Type of Service** annotations is to understand the function of websites, and how they might be related to the function of real user conversations with general-purpose models trained on this web data. This is distinct from the text pretraining domain analysis conducted in prior work [70], as the annotations are not about the relevant source or topic (e.g. legal, biomedical, social, etc), but the functional purpose of the website for users. The taxonomy was developed after authors reviewed hundreds of websites themselves, compared categories, and clustered common functions.

**Instructions for "Type of Service"**

What is the purpose or service of the website? This is relevant to US copyright infringement analysis into the "effect of the use on the potential market for or value of the work". i.e. will copying this data jeopardize the website's business.
We have listed out some common types of service below. Using the "website description" you wrote, pick the best fitting type of service, or if none of these fit exactly, write your own (Other) e.g. "Video Game Blogging". We will later create more clusters based off these suggestions. Here are the starter options:

- Ecommerce (e.g. Amazon, gaming, etc)
- Periodicals (News, magazine) (e.g. NYTimes, LATimes, Forbes, etc)

- Social Media (e.g. Twitter, Facebook, Reddit, etc)
- Encyclopedia/Database (e.g. Wikipedia, IMDB, etc)
- Academic (e.g. pubmed, nature, journals.plos.org, etc)
- Government (e.g. sec.gov, justia.com, parliament.uk, etc)
- Company/Organization/Personal website (e.g., www.ge.com)
- Blog websites (e.g., www.medium.com)
- Other: In a second stage, we will expand the list above

The purpose of annotating for **Sensitive Content** is to understand the distribution of content that practitioners may wish to exclude from their corpus for reasons of toxicity, bias, nudity, hate speech, or other offensive topics.

---

**Instructions for "Illegal/Sensitive/NSFW Content"**

Does the website contain a non-negligible amount of pornography, drug content, violence, promotion of illegal activities, or hate speech. This should only be yes, if it's more than a minimal amount, for example while there are some sensitive things in Wikipedia, the answer is no; whereas the answer is yes for Reddit.
Options:

- Pornography: y/n
- Drug content : y/n
- Violence: y/n
- Promotion of illegal activities: y/n
- Hate speech: y/n

---

### C.2.2 Pretraining Datasets (Crowdworker)

**General instructions**

Please read the below instructions carefully, as accuracy is crucial for our analysis, and the choices are sometimes nuanced. Turn off your ad blockers or browser extensions for this task. Inspect each website thoroughly, navigating through many pages. This is essential for finding ads, paywalls, videos, and audio content that may not be on the main page of the website.

---

**Instructions for Website Issue**

Some websites have been sold or shut down since the scrape. In these cases, check the box for website issues and don't continue.

---

**Instructions to Annotate "Terms of Service Link(s)"**

For each website domain, we want to find all links that are related to the domain's terms, including around general use, data, content, privacy, etc.. This will allow us to later identify all legal terms associated with using the website, its content or data. It is critically important that main terms pages are not missed, so we will randomly review some to make sure we are getting a comprehensive list. The most important policies for our work are copyright-related policies.

Here are 3 examples of the terms found for a website:

**imdb.com** Links:

- `https://www.imdb.com/conditions`
- `https://www.imdb.com/licensing/subservicetc/`
- `https://www.imdb.com/privacy`

**plos.org** Links:

- `https://plos.org/terms-of-service/`
- `https://plos.org/text-and-data-mining/`
- `https://plos.org/terms-of-use/`
- `https://plos.org/privacy-policy/`

**goodreads.com** Links:

- `https://www.goodreads.com/about/terms`
- `https://www.goodreads.com/about/privacy`
- `https://www.goodreads.com/api/terms`

Suggested procedure to find the links:

1. Many websites have links to their terms, privacy, or content policies at the bottom of their main page. Scroll to the very bottom and see if any exist.

2. Sometimes not all relevant terms will appear there. We recommend you also search for:

    (a) "<website name> terms of use"
    (b) "<website name> copyright policy"
    (c) "<website name> content policy"
    (d) "<website name> privacy policy"
    (e) "<website name> developer policy"
    (f) "<website name> data mining"

3. ONLY include pages you find that appear to be relevant to the legal conditions/terms of using the website or data in some capacity. Very rarely, websites may have hundreds of these pages. In those cases, feel free to just include the top few main ones.

## Instructions to Annotate "Paywall"

Does the website paywall any of its content? We hope to see what websites require some sort of paid subscription or sign up (even if it offers free starter trials) in order to view their content.
Output options:

- No – we did not find any paywall for any of the content. Examples: Wikipedia, Reddit, Youtube.

- Some – a fair amount of content can be viewed without any issue (e.g. multiple news articles), but after some reading/searching there appears to be a paywall on the rest of the content. Examples: https://www.popularmechanics.com/.

- All – every main page of content is paywalled. This means that no single webpage or article of content can be fully read without subscribing in some way. Examples: NYTimes, Wall Street Journal.

Suggested procedure to determine if there is a paywall:

1. Make sure you are not logged into any accounts on your browser, especially ones applicable to the website.

2. Explore the website content and see if a paywall request appears.

3. Double check by searching: "does <website name> have a paywall?"

## Instructions to Annotate "Content Modalities"

What modalities of content appear on the website? A modality is the actual content of the website, for which we have four options: text, images, videos, audio. These modalities can appear at different levels, depending on the website. Do not count the content in automatic embedded advertisements towards this.

- For text, there must be at least one paragraph or multiple sentences/captions on the website.
- For images, there must be at least one or more distinct images embedded on the page. Visual styling that is part of the website design does not count.
- For videos, there must be at least one embedded video – often they are not on the main page, so you may need to look.

Output options:

- Text
- Images
- Videos
- Audio

Levels of modality appearing on the website:

- No – Content of this type is not on the website.
- Yes – There is content of this type, even if it's not common, like images on Wikipedia. Do not count visual styling/illustrations that are just part of the natural website design – the presence of image(s) should be notable. Do not count the content in ads.

Suggested procedure:

1. Try to find representative webpages on the website; if there is a search bar try to search for some generic terms
2. Explore enough pages to be able to make a confident assessment of how much of each modality is present.

## Instructions to Annotate "Advertisements"

Do third-party advertisements appear on the website? Many websites host advertisements to make money. They may appear on the top, bottom, or side bars of just some pages, so look thoroughly. Self promotion does not count. These may not be on the main website page. Remember to turn off your ad blockers / extensions.
Output options:

- No – No automatic advertisements are integrated into the pages.
- Yes – Some automatic advertisements do appear on the pages.

Suggested procedure:

1. Search through the website and its content, looking for advertisements.

# D   Automatic Annotation Methodology Details

## D.1   Robots.txt Taxonomy

Using the Wayback Machine, we snapshotted websites' robots.txt and terms of service at monthly intervals from January 2016 to April 2024. For each web domain, we identified scraping constraints for the wildcard ("*") as well as the user agents of the the six organizations commonly known to train AI models (Google, OpenAI, Anthropic, Cohere, Meta, Common Crawl). See Table 5 for details on each of these organizations.

We then categorized the robots.txt restrictions for every web domain across an ascending spectrum of restrictions. These were:

1. No robots.txt present.

2. No restrictions or sitemap: a simple directive allowing unrestricted access to crawlers, e.g.

   ```
   User-agent: *
   Disallow:
   ```

3. Only a sitemap is present: a list of all URLs on the website along with metadata, helping search engines index the site more thouroughly and efficiently.

4. Only a sitemap and crawl delay are present: Limit the frequency of crawler requests to the server, often included to prevent a site from being overloaded with too many requests- this affects the crawling rate but not accessibility.

5. Search and query restrictions apply: disallow directives that match patterns associated with search result pages or URLs containing query parameters, e.g.

   ```
   Disallow: /search
   Disallow: /*?*
   ```

6. Crawling specific directories is prohibited: many sites have confidential or private directories that should not be crawled

7. Agent is fully disallowed from crawling any parts of the website

## D.2   Robots.txt Agents

In Table 5 we detail the AI-related organizations, their agents and their accompanying documentation, where present. In Table 6 we show the statistics for agents across all the robots.txt we analyzed. Lastly, Figure 5 describes the observed company-to-company conditional probabilities for robots.txt restrictions, to understand how agent restrictions are prioritized among many web administrators.

| Organization | User Agent | Details | Docs | Purpose |
|---|---|---|---|---|
| OpenAI | GPTBot | OpenAI's official user-agent for crawling and collecting training data from the web. | ● | Training |
| | ChatGPT-User | OpenAI's official user-agent for live user queries that trigger browsing plugins. According to OpenAI's documentation, their current opt-out implementation treats both user agents the same. | ● | Retrieval |
| Google | Google-Extended | Google's official user-agent for "Gemini Apps, Vertex AI generative APIs, and future generations of models." | ● | Training |
| Google Search | Googlebot | Google's official user-agents for general web crawling, related to their search engine. | ● | Web Search |
| Common Crawl | CCBot | Common Crawl's user-agent for maintaining open access archives of the web, particularly for research. | ● | Archive, research |
| Anthropic | ClaudeBot | Anthropic's official user-agent for crawling and collecting training data from the web. Their policy statest that their opt-outs respect this agent as well as Common Crawl's CCBot. | ● | Training, retrieval |
| False Anthropic | anthropic-ai | An unofficial but widely adopted user-agent, presumably to disallow any Anthropic data crawling and collection. | ○ | Training |
| | Claude-Web | An unofficial but widely adopted user-agent, presumably for live queries in Claude which trigger browsing. | ○ | Retrieval |
| Meta | FacebookBot | Meta's official user-agent for crawling and collecting training data from the web | ● | Training, retrieval |
| Cohere | cohere-ai | An unofficial but widely adopted user-agent, presumably to disallow any Cohere data crawling and collection. | ○ | Training, retrieval |
| Internet Archive | ia_archiver | The official user-agent that supports the Wayback Machine open web archive. The Internet Archive may ignore this user-agent | | Training, retrieval |
| *All Agents* | All | Notation for our aggregation of robots.txt policies towards all agents. This is used to track if a website is fully lenient or restrictive to all user agents. | | |

Table 5: The **list of organizations we trace, and their associated web crawler user-agents.** We provide basic details on these crawlers, and links to their documentation where provided. We list the stated purpose of crawlers, including for Training AI models, for Retrieving relevant information for a general-purpose AI system, for conducting Web Search, or for creating an Archive or the web.

Figure 5: We compute the percentage that organization B is restricted by a web domain's robots.txt, given organization A's agents have been restricted. The organizations include AI companies (OpenAI, Google, Anthropic, Anthropic's False agents, Cohere, Meta), non-profit web archives (Common Crawl, the Internet Archive), and then a general web search agent (Google Search). **We find OpenAI web agents are nearly always disallowed if any AI organizations are disallowed, but the reciprocal is less frequent.**

| Agent Name | # Observed | All Disallowed | | Some Disallowed | | None Disallowed | |
|---|---|---|---|---|---|---|---|
| | | Count | % | Count | % | Count | % |
| *All Agents* | 269,212 | 1175 | 0.44% | 198642 | 73.79% | 69395 | 25.78% |
| * | 226,903 | 2935 | 1.29% | 183364 | 80.81% | 40604 | 17.89% |
| Mediapartners-Google | 20,848 | 749 | 3.59% | 3710 | 17.80% | 16389 | 78.61% |
| Googlebot | 12,831 | 44 | 0.34% | 9544 | 74.38% | 3243 | 25.27% |
| MJ12bot | 10,556 | 5962 | 56.48% | 319 | 3.02% | 4275 | 40.50% |
| Twitterbot | 10,385 | 29 | 0.28% | 3413 | 32.86% | 6943 | 66.86% |
| Slurp | 10,070 | 507 | 5.03% | 4364 | 43.34% | 5199 | 51.63% |
| AhrefsBot | 9,824 | 5506 | 56.05% | 1516 | 15.43% | 2802 | 28.52% |
| IRLbot | 9,142 | 134 | 1.47% | 157 | 1.72% | 8851 | 96.82% |
| Yandex | 8,948 | 2901 | 32.42% | 1623 | 18.14% | 4424 | 49.44% |
| bingbot | 8,077 | 195 | 2.41% | 3079 | 38.12% | 4803 | 59.47% |
| Googlebot-News | 7,958 | 82 | 1.03% | 7548 | 94.85% | 328 | 4.12% |
| Baiduspider | 7,789 | 3476 | 44.63% | 1933 | 24.82% | 2380 | 30.56% |
| msnbot | 6,784 | 261 | 3.85% | 2737 | 40.34% | 3786 | 55.81% |
| 008 | 5,661 | 5360 | 94.68% | 108 | 1.91% | 193 | 3.41% |
| ia_archiver | 5,604 | 2768 | 49.39% | 1822 | 32.51% | 1014 | 18.09% |
| SemrushBot | 5,418 | 3865 | 71.34% | 84 | 1.55% | 1469 | 27.11% |
| Googlebot-Image | 5,082 | 728 | 14.33% | 1842 | 36.25% | 2512 | 49.43% |
| Nutch | 5,011 | 4446 | 88.72% | 0 | 0.00% | 565 | 11.28% |
| ccBot | 5,005 | 3227 | 64.48% | 1328 | 26.53% | 450 | 8.99% |
| facebookexternalhit | 2,286 | 31 | 1.36% | 323 | 14.13% | 1932 | 84.51% |
| NPBot | 2,261 | 2259 | 99.91% | 0 | 0.00% | 2 | 0.09% |
| wget | 2,234 | 2234 | 100.00% | 0 | 0.00% | 0 | 0.00% |
| rogerbot | 2,147 | 736 | 34.28% | 703 | 32.74% | 708 | 32.98% |
| libwww | 2,138 | 2046 | 95.70% | 0 | 0.00% | 92 | 4.30% |
| SemrushBot-SA | 2,108 | 1428 | 67.74% | 10 | 0.47% | 670 | 31.78% |
| sitecheck.internetseer.com | 2,064 | 1972 | 95.54% | 0 | 0.00% | 92 | 4.46% |
| Download Ninja | 2,060 | 1971 | 95.68% | 0 | 0.00% | 89 | 4.32% |
| ZyBORG | 2,059 | 1941 | 94.27% | 0 | 0.00% | 118 | 5.73% |
| Zealbot | 2,058 | 1969 | 95.68% | 0 | 0.00% | 89 | 4.32% |
| Xenu | 2,048 | 1959 | 95.65% | 0 | 0.00% | 89 | 4.35% |
| Facebot | 2,020 | 0 | 0.00% | 936 | 46.34% | 1084 | 53.66% |
| linko | 1,991 | 1902 | 95.53% | 0 | 0.00% | 89 | 4.47% |
| ChatGPT-User | 895 | 750 | 83.80% | 118 | 13.18% | 27 | 3.02% |
| anthropic-ai | 260 | 229 | 88.08% | 1 | 0.38% | 30 | 11.54% |
| cohere-ai | 185 | 180 | 97.30% | 1 | 0.54% | 4 | 2.16% |
| Google-Extended | 871 | 836 | 95.98% | 4 | 0.46% | 31 | 3.56% |
| Amazonbot | 546 | 358 | 65.57% | 109 | 19.96% | 79 | 14.47% |
| FacebookBot | 235 | 220 | 93.62% | 2 | 0.85% | 13 | 5.53% |
| ClaudeBot | 45 | 40 | 88.89% | 0 | 0.00% | 5 | 11.11% |
| Claude-Web | 89 | 82 | 92.13% | 0 | 0.00% | 7 | 7.87% |

Table 6: **A breakdown of the top 60 web crawler agents mentioned across the robots.txt for all 14k web domains we analyzed.** Those highlighted in gray are related to the organizations in our analysis (see a detailed summary of them in Table 5). We compute the number of times each agent is observed, as well as the proportion of times a robots.txt restricts it either fully, partially, or not at all. Lastly, the *All Agents* row refers to the number of total observations, as well as the tally of instances where a robots.txt fully restricts every agent, partially restricts every agent, or restricts no agents at all.

### D.3 Terms of Service Taxonomy

After a close reading of hundreds of ToS pages, the paper authors noted three distinct indicators for metered data usage: competing service clauses, license type, and in some cases, explicit crawling and AI policies. To identify clauses relating to these topics at scale, we utilized the *GPT-4o* model with custom prompting, sending requests through the OpenAI API. This section will detail the taxonomies we developed for categorizing the ToS pages, as well as the prompt engineering and annotation methodology behind automating the process.

Our taxonomies were designed with the variant nature of legal documents in mind. While we initially tried to categorize ToS pages as either, *TRUE or FALSE*, for containing a policy relating to the taxonomy at hand, we quickly found examples that broke this mold. In order to account for nuanced clauses, our final taxonomies consist of multiple categories in ascending order of restrictiveness. The order and definitions were refined as we came across enough additional examples to demand their own category. For our temporal analysis, this structure allows us to better express the tightening restrictions on web data over time.

See the finalized taxonomies below:

---

**Competing services taxonomy**

1. **Non-Compete**
   - **Definition**: the ToS includes a clause that specifically prohibits the use of its content, data, or materials for competing services. This category relates to commercialization or other commercial uses of the site's content and does not include clauses that solely restrict scraping, storing data, or distributing data.

2. **No Re-Distribution**
   - **Definition**: the ToS prohibits the distribution or reselling of content. This includes clauses restricting selling content or creating and distributing datasets. Does not include general commercial usage restrictions unless they directly pertain to redistribution.

3. **Non-Compete/No Re-Distribution**
   - **Definition**: both of the above categories are present in the given ToS.

4. **No restrictions**
   - **Definition**: the ToS does not include clauses that restrict competing services or re-distribution.

---

**License type taxonomy**

1. **Personal/Noncommercial/Research Only**
   - **Definition**: the ToS explicitly states that the content is available for personal, noncommercial, or research purposes only. Commercial use of the content is strictly prohibited.

2. **Conditional Commercial Access**
   - **Definition**: the ToS specifies that only certain parts of the website are open-access or commercially viable, while other parts are restricted. Commercial use is allowed under specific conditions (for example: permission must be granted for commercial purposes, Commercial use is allowed but third-party reposting is prohibited, non-compete clauses restrict using the content in ways that compete with the service provider).

3. **Open or Unrestricted Commercial Use**

---

> - **Definition**: the ToS does not explicitly disallow commercial use, indicating that the website content is open for use or considered public information. This category includes terms that allow commercial use without specific restrictions or conditions (for example: creative Commons licenses permit commercial use).

**Crawling and AI taxonomy**

1. **Prohibits crawling and AI unconditionally**
   - **Definition**: the ToS explicitly states that both crawling and the use of data for AI or ML are prohibited without exception.
2. **Prohibits crawling unconditionally, but no mention of AI**
   - **Definition**: the ToS explicitly states that crawling and associated activities (such as to copy, use, or distribute and other automated means) are prohibited with no exceptions or conditions. Does not mention any restrictions to AI or ML uses.
3. **Prohibits AI unconditionally, but not crawling**
   - **Definition**: the ToS explicitly prohibits AI or ML usage without exception but doesn't mention a policy on crawling.
4. **Only prohibits crawling and AI under certain conditions, or to certain parts of the website**
   - **Definition**: the ToS provides conditions under which crawling and the use of data for AI or ML are restricted or permitted. This category includes clauses containing "With the exception of material marked 'Open Access'...".
5. **No restrictions on crawling or AI**
   - **Definition**: the ToS does not contain any clauses or mentions regarding the prohibition or restriction of crawling or the use of data for AI or ML. Both activities are implicitly allowed.

## D.4  Prompt Engineering

For each indicator—competing services, license type, and crawling and AI policies—we developed a unique prompt directing *GPT-4o* to produce a verdict and supply directly quoted evidence. Based on our taxonomy, the verdict corresponds to the best fitting category and the evidence is each instance of text contributing to said verdict for a given ToS.

Our prompts were refined through an iterative process, comparing each output with a "gold answer" annotation set to uncover shortcomings and improve (see section Appendix D.5 for details on the annotation process). For each version of our prompt, we analyzed false negatives (clauses that the model failed to recognize) to widen our scope and improve the specificity of our prompt, and analyzed false positives (clauses that the model recognized incorrectly) to narrow our scope. The prompts were modified using this methodology until preforming with an average accuracy of 85% or higher against our annotation set [table 7].

See the finalized prompts below:

**Competing services prompt**

Your task is to analyze the provided Terms of Service (ToS) document to determine if there are specific restrictions related to competing services or the redistribution of content. You will categorize each ToS based on the following taxonomy:

1. Non-Compete:
   Definition: the ToS includes a clause that specifically prohibits using or sharing its content or data to create competing services. This does not include clauses solely restricting scraping, storing data, or non-commercial use.

2. No Re-Distribution:
   Definition: the ToS prohibits the distribution or reselling of content. This does not include general commercial usage restrictions unless they directly pertain to redistribution.

3. Non-Compete and No Re-Distribution:
   Definition: both of the above categories are present in the given ToS.

4. No restrictions:
   Definition: the ToS does not include clauses that restrict competing services or re-distribution.

Return ONLY a dictionary with your verdict (a category number from the taxonomy) and the corresponding evidence. Evidence does not need to be continuous, you should include all mentions of a non-compete or no-redistribution. Do NOT include any additional text in your response do NOT wrap your response with "'json"'. Format the response exactly like these examples:

```
- {"verdict":1, "evidence": "Exact text from ToS detailing
Non-Compete."}
- {"verdict": 2, "evidence": "Exact text from ToS detailing
No Re-Distribution."}
- {"verdict": 3, "evidence": "Exact text from ToS detailing
Non-Compete; Exact text from ToS detailing No Re-Distribution"}
- {"verdict": 4, "evidence": "N/A"}
```

This format will assist in a comprehensive review of the ToS and allow for accurate categorization based on the specific language used in the document.

---

**License type prompt**

Your task is to analyze the provided Terms of Service (ToS) document to determine the license type, categorizing it based on the following taxonomy. Use direct quotes from the ToS as evidence to support your categorization. Focus on the explicit language used regarding permissions and restrictions for personal, noncommercial, or commercial use.

1. Personal/Noncommercial/Research Only:
   Definition: the ToS restricts use to personal, noncommercial, or research purposes without any exceptions allowing commercial use.

2. Conditional Commercial Access:
   Definition: the ToS contains restrictions on general use but specifies conditions under which commercial use is permitted. This includes needing permissions, complying with certain conditions, or paying fees for commercial use. Look for terms like 'requires written permission,' 'subject to approval,' or 'commercial use permitted under conditions.'

3. Open or Unrestricted Commercial Use:
   Definition: the ToS permits commercial use broadly, without requiring additional permissions or adhering to specific conditions. This includes terms that explicitly allow or imply commercial use is permitted across all contents.

Return ONLY a dictionary with your verdict (a category number from the taxonomy) and the corresponding evidence. Evidence does not need to be continuous, you should include all

mentions of a license type. Do NOT include any additional text in your response do NOT wrap your response with "'json"'. Format the response exactly like these examples:

```
    - {"verdict": 1, "evidence": "Exact text from ToS detailing
Personal/Noncommercial/Research Only license."}
    - {"verdict": 2, "evidence ":" Exact text from ToS detailing
 Conditional Commercial Access license."}
    - {"verdict": 3, "evidence ": "Exact text from ToS detailing
Open or Unrestricted Commercial Use license or 'N/A' if there is no
explicit mention."}
```

This will assist in a comprehensive review of the ToS and allow for accurate categorization based on the specific language used in the document. Ensure that your assessment is detailed and directly references the ToS document.

---

### Crawling and AI prompt

Your job is to analyze the Terms of Service (ToS) document that I will provide you to determine the policy on web scraping and artificial intelligence (AI) or machine learning (ML). You will categorize each ToS document based on the following taxonomy:

1. Prohibits scraping and AI unconditionally:
   Definition: the ToS explicitly states that both scraping and the use of data for AI or ML are prohibited without exception.

2. Prohibits scraping unconditionally, but no mention of AI:
   Definition: the ToS explicitly states that scraping and associated activities (such as to copy, use, or distribute and other automated means) are prohibited with no exceptions or conditions. Does not mention any restrictions to AI or ML uses.

3. Prohibits AI unconditionally, but not scraping:
   Definition: the ToS explicitly prohibits AI or ML usage without exception but doesn't mention a policy on scraping.

4. Only restricts or permits scraping and AI under certain conditions, or to certain parts of the website:
   Definition: the ToS provides conditions under which scraping and the use of data for AI or ML are restricted or permitted. This category includes clauses containing "With the exception of material marked 'Open Access'...".

5. No restrictions on scraping or AI:
   Definition: the ToS does not contain any clauses or mentions regarding the prohibition or restriction of scraping or the use of data for AI or ML. Both activities are implicitly allowed.

Return ONLY a dictionary with your verdict (a category number from the taxonomy) and the corresponding evidence. Evidence does not need to be continuous, you should include all mentions of a scraping, AI or ML policy. Do NOT include any additional text in your response do NOT wrap your response with "'json"'. Format the response exactly like these examples:

```
- {"verdict": 1, "evidence": "Exact text from  ToS detailing explicit
scraping AND AI      prohibition."}
- {"verdict": 2, "evidence": "Exact text from  ToS detailing explicit
scraping prohibition."}
- {"verdict": 3, "evidence": "Exact text from  ToS detailing explicit
AI prohibition"}
```

```
- {"verdict": 4, "evidence": "Exact text from  ToS detailing the
  conditions restricting scraping and or AI."}
- {"verdict": 5, "evidence": "N/A"}
```

This will assist in a comprehensive review of the ToS and allow for accurate categorization based on the specific language used in the document. Ensure that your assessment is detailed and directly references the ToS document.

## D.5 Annotating and Scoring

To empirically measure the ability of *GPT-4o* to follow our prompts and taxonomies, we manually audited a sample of 100 URLs for each indicator in question: competing services, license type, and crawling and AI. The URLs were randomly sampled from the 10K Random Subset and each ToS link was carefully reviewed for clauses relating to our taxonomies. We found that most of the relevant clauses were located in the Terms of Service (rather than a Privacy Policy or Copyright Notices page), and we saved all verdicts and corresponding evidence to our annotation dataset for comparison with the results from *GPT-4o*. With this "gold answer" annotation set, we calculated the mico-average precision/recall of each prompt to ensure all class labels from our taxonomy were weighted relative to their size, and the results are reported in Table 7.

| Scoring metric | Competing Services | License Type | Crawling and AI Policy |
|:---:|:---:|:---:|:---:|
| Precision/Recall | 0.92 | 0.85 | 0.89 |

Table 7: Precision and Recall Values for each prompt against the annotation set. Each score is a mico-average of all the individual class scores.

## D.6 WildChat Annotation

In Figure 4, we distinguish a wide range of different types of service related user prompts that serve various purposes:

- **Creative Composition:** These requests involve role-playing, fictional story writing, or continuing existing narratives, allowing users to explore their imaginative capabilities.
- **Academic Composition:** These focus on non-fiction essay writing, continuation, or editing, aiding in scholarly and professional writing.
- **Coding composition:** These requests ask for assistance fixing, debugging, or general coding help, supporting developers.
- **Brainstorming, planning, or ideation:** These requests ask the system to help brainstorm, generate ideas, or plan out a project.
- **Explanation & Reasoning:** These prompts ask the system to explain or reason through a question, help with puzzles, math problems or other problem-solving tasks.
- **Self-help:** These requests seek advice or support for personal issues, providing a platform for guidance.
- **Sexual content:** These requests are related to sexually explicit content requests—such as sexual role-play or fiction.
- **News:** These prompts request information related to news, recent events, are generally current affairs that may be applicable to news websites.
- **E-commerce Information:** These requests inquire about products and purchasing information.
- **Translation:** These requests ask for aid in translating text from one language to another, assisting users in overcoming language barriers.

- **Organization Information:** These requests ask for information specific to organizations, companies, or individuals, which may pertain to organization/personal websites.

To assess the accuracy of using GPT-4o's service type predictions, we conducted a manual evaluation of 50 randomly sampled WildChat prompts. Each prompt's predicted type of service was reviewed for correctness, resulting in an error rate of 18%. The system prompt for GPT-4o is shown below:

---

**System Prompt Used for WildChat Analysis**

You are a categorization assistant. I will provide you with a user prompt and a response. Your task is to classify the prompt into one or more of the following *'Type of Service'* categories.

**Categories:**
- General informational requests
- Creative composition
- Academic composition
- Coding composition
- Brainstorming, planning, or ideation
- Asking for an explanation, reasoning, or help solving a puzzle or math problem
- Translation
- Self-help, advice seeking, or self-harm
- Sexual or sexual roleplay content requests
- News or recent events informational requests
- E-commerce or information requests about products and purchasing
- Information requests specifically about organizations, companies, or persons
- Other (choose this only as a last resort)

**Descriptions (do not include these in the labels):**
- Creative composition: such as role-playing, fictional story writing, or continuation
- Academic composition: such as non-fiction essay writing, continuation, or editing
- Coding composition: fixing, debugging, or help
- Brainstorming, planning, or ideation
- Asking for an explanation, reasoning, or help solving a puzzle or math problem
- Self-help: advice seeking, or self-harm
- Sexual or illegal content requests: inappropriate or illicit content requests
- News or recent events informational requests
- E-commerce: information requests about products and purchasing
- Information requests: specifically about organizations, companies, or persons

Provide the classification in the following JSON format:

```
{
    "Type of Service": []
}
```

---

## E  Wayback Machine

Our temporal data collection and processing pipeline consists of three main components: (1) a Wayback Machine client for retrieving historical web snapshots; (2) a file processing system for

extracting and formatting textual content; and (3) a temporal analysis module for tracking changes over time. The implementation was done in Python, utilizing concurrent processing for efficiency.

## E.1   Data Collection

We developed a custom client that interfaces with the Internet Archive's Wayback Machine CDX API to retrieve historical snapshots of web pages. The client implements a rate-limiting decorator that enforces a maximum of two requests per second using a sleep-and-retry mechanism. This approach ensures consistent API access while preventing overload of the Wayback Machine's servers. When rate limits are exceeded, the system automatically pauses and retries the request after an appropriate delay.

The client employs multi-threading to process multiple URLs concurrently. For each URL, the system queries the CDX API to identify available snapshots within the specified date range, filters snapshots based on the desired frequency (daily, monthly, or annual), excludes duplicate content using digest-based deduplication, and downloads and stores the HTML content of unique snapshots. In order to save memory, we skip saving snapshots if the snapshot contents have not changed since the previous snapshot.

## E.2   Content Processing

The pipeline processes the collected HTML snapshots through several stages. The text extraction phase utilizes BeautifulSoup4 to parse HTML and extract meaningful text content. During content formatting, the system preserves document structure (headings, paragraphs), removes scripting and styling elements, maintains hierarchical formatting for better readability, and handles various text encodings using chardet for robust character detection.

For data organization, the system implements a structured storage format. Each snapshot is stored in a JSON structure where the top level consists of domain keys mapping to nested dictionaries. These nested dictionaries contain timestamp-indexed snapshots, with each snapshot containing the processed text content and associated metadata. The output format follows this structure:

```
{
    "domain.com": {
        "terms_url_1": {
            "2023-01-15": "processed text content...",
            "2023-07-15": "processed text content..."
        },
        "terms_url_2": {
            "2023-02-01": "processed text content...",
            "2023-08-01": "processed text content..."
        }
    }
}
```

For large datasets, the system implements automatic chunking, creating multiple JSON files when the data size exceeds a configurable threshold. Each chunk maintains the same structural format while ensuring memory efficiency.

## E.3   Implementation Details

The implementation leverages both multi-threading and multi-processing for parallel execution. ThreadPoolExecutor manages concurrent API requests and file processing, while multiprocessing handles CPU-intensive text extraction tasks. The system implements comprehensive error logging and failed URL tracking, storing detailed error information for failed requests.

Memory management is handled through chunked processing for large datasets, with configurable chunk sizes to accommodate varying system capabilities. The processed data is stored in JSON format, with automatic file splitting when size thresholds are exceeded. The entire pipeline accepts command-line arguments for configuring parameters such as date ranges, snapshot frequency, number of worker threads, and output formats.

### E.4 Limitations

Reliance on the Internet Archive's Wayback Machine introduces several inherent limitations. First, the archive's crawl frequency is inconsistent across websites and time periods, potentially missing important changes between snapshots. This non-uniform temporal coverage varies significantly based on a site's popularity, with less frequently visited sites having larger gaps between snapshots. Additionally, some snapshots may be incomplete due to robots.txt restrictions, JavaScript-dependent content, or crawl errors, affecting our ability to capture the complete state of a webpage at a given time.

## F Forecasting

### F.1 Methodology

Seasonal AutoRegressive Integrated Moving Average (SARIMA) is a widely used statistical method for time series forecasting that captures both trend and seasonal patterns [91, 92]. Denoted as $SARIMA(p, d, q)(P, D, Q)_m$, it extends the ARIMA model by incorporating seasonal components [93, 94]. The parameters $p$, $d$, and $q$ represent the order of autoregression, degree of differencing, and order of moving avergae for the non-seasonal part, while $P$, $D$, and $Q$ represent their seasonal counterparts, with $m$ indicating the number of periods per season [95, 96]. SARIMA has been successfully applied in various domains, including economics [97], energy [98], and environmental studies [99]. The model's effectiveness lies in its ability to account for autocorrelation, trend, and seasonality in time series data [100, 101]. SARIMA assumes that the time series is stationary (or can be made stationary through differencing) and that the errors are uncorrealted and have zero mean [102]. While it can capture a wide range of time series behaviors, it may not be suitable for series with complex nonlinear dynamics or those exhibiting heteroscedasticity [103, 104]. SARIMA does not account for sudden or altering events, implicitly relying on the assumption that current patterns will continue in some form, which can limit its effectiveness.

We chose the SARIMA parameters through an automated model selection process using the `auto.arima` function of the `pmdarima` package [105], and fit the models with the statsmodels implementation of SARIMA [106]. The automated selection process chooses lag and differencing orders, along with other parameters, to optimize the Akaike information criterion. We show the selected parameters and their interpretations in Table 8.

Table 8: SARIMA parameter interpretation

| Parameter | Value | Description |
|---|---|---|
| Non-seasonal order | (2, 1, 2) | - AutoRegressive (AR) order: 2 |
| | | - Integrated (I) order: 1 |
| | | - Moving Average (MA) order: 2 |
| Seasonal order | (1, 1, 1, 6) | - Seasonal AutoRegressive (SAR) order: 1 |
| | | - Seasonal Integrated (SI) order: 1 |
| | | - Seasonal Moving Average (SMA) order: 1 |
| | | - Seasonal periodicity: 6 |

The non-seasonal order (2, 1, 2) indicates that the current value being predicted is dependent on the past 2 observations, the time series is differenced once to achieve stationarity, and the prediction is influenced by the past 2 forecast errors. Similarly, the seasonal order (1, 1, 1, 6) indicates that the current value is affected by the previous seasonal value, the seasonal component is differenced once to remove seasonal non-stationarity, the prediction is impacted by the past seasonal forecast error, and the data exhibits a recurring pattern every 6 periods.

### F.2 Limitations

The restriction trends are forecast a year into the future only to provide a short-term sense of how restrictions might evolve, in the absence of significant exogenous factors. We caution the reader that this is a very strong assumption, as the outcomes of lawsuits, changing company practices, and the community's response could all have significant effects on the restrictions applied to data. These

Table 9: Coefficients and associated hypothesis tests for SARIMA models on the share of tokens restricted, over all three corpora.

(a) C4.

|  | coef | std err | z | P>\|z\| | [0.025 0.975] |
|---|---|---|---|---|---|
| ar.L1 | -0.5980 | 0.231 | -2.584 | 0.010 | -1.052 -0.144 |
| ma.L1 | 0.7518 | 0.285 | 2.639 | 0.008 | 0.193 1.310 |
| ar.S.L2 | -0.1074 | 0.191 | -0.562 | 0.574 | -0.482 0.267 |
| ma.S.L2 | -0.4791 | 0.227 | -2.108 | 0.035 | -0.924 -0.034 |
| sigma2 | 1.943e-05 | 1.39e-06 | 13.934 | 0.000 | 1.67e-05 2.22e-05 |

(b) Dolma.

|  | coef | std err | z | P>\|z\| | [0.025 0.975] |
|---|---|---|---|---|---|
| ar.L1 | -0.8510 | 0.236 | -3.602 | 0.000 | -1.314 -0.388 |
| ma.L1 | 0.7399 | 0.276 | 2.658 | 0.008 | 0.199 1.280 |
| ar.S.L6 | -0.0747 | 0.183 | -0.408 | 0.683 | -0.434 0.284 |
| ma.S.L6 | -0.9154 | 0.298 | -4.398 | 0.000 | -1.323 0.507 |
| sigma2 | 2.521e-05 | 4.069e-06 | 6.198 | 0.000 | 1.71e-05 3.33e-05 |

(c) RefinedWeb.

|  | coef | std err | z | P>\|z\| | [0.025 0.975] |
|---|---|---|---|---|---|
| ar.L1 | 0.1996 | 0.156 | 1.280 | 0.201 | -0.106 0.505 |
| ma.L1 | -0.9145 | 0.124 | -7.367 | 0.000 | -1.158 -0.671 |
| ar.S.L16 | -0.1649 | 0.122 | -1.354 | 0.176 | -0.403 0.074 |
| ma.S.L16 | -0.6390 | 0.159 | -4.025 | 0.000 | -0.950 -0.328 |
| sigma2 | 6.146e-05 | 4.43e-06 | 13.871 | 0.000 | 5.28e-05 7.01e-05 |

forecasts and trends are also most relevant to large, data-intensive, general-purpose models as they exist now; increased use of smaller, more specialized or less data-dependent models may reduce the relevance of the trends we identify. We accordingly focus our analysis on current statistics, and point to Appendix F.1 for supporting information on SARIMA and backtests of our fitted model.

# G   Extended Related Work

**Data Documentation**   Previous work has highlighted the importance of data documentation in machine learning [54, 71, 72, 73, 74]. These works particularly stress the challenges posed by poor documentation to reproducibility, sound scientific practice, and understanding of model behavior [68, 69, 70]. Recent research has also explored the significance of documenting AI ecosystems [107] and the supply chain from data to models [108]. Previous studies have strongly advocated for and provided frameworks for documentation and audits to enhance transparency and accountability in AI systems [109, 110, 111]. Similar to Longpre et al. [11] which leverages the collective knowledge of legal and machine learning experts, earlier research has emphasized the importance of interdisciplinary collaborations [112]. 'Datasheets for Datasets' [66] and 'Data Statements' [67] both offer structured frameworks for revealing essential metadata, such as the motivation behind and intended use of datasets. Pushkarna et al. [113] expanded on datasheets with 'Data Cards' that include sources, collection methods, ethics, and adoption information. Additionally, Mitchell et al. [114] introduced model cards to benchmark model performance across demographic groups and disclose evaluation procedures. Crisan et al. [115] proposed interactive model cards as an alternative mode of documentation and metadata sharing. Complementary to transparency regarding the dataset creation process, Corry et al. [116] provides a framework to guide users on how to navigate datasets as they approach the end of their life cycle. Longpre et al. [75] highlighted the importance of building standards across data documentation, aspects of which we draw on here.

**Data Audits**   Several works stressed the importance of auditing datasets used to train models [38, 50, 12]. Building these audits into the workflow on AI is critical to building accountability [117, 118]. While previous works audited a variety of datasets and modalities, our work presents the largest and most comprehensive analysis of data used in training foundation models. This is becoming more pressing with the increasing richness of data sources, including those compiled by academics [78, 119, 120], synthetically generated by models [121, 122], or aggregated by platforms like Hugging Face [123]. The trend of combining and re-packaging numerous datasets and web sources has become prevalent among practitioners [124, 125, 126, 127]. Significant work has gone into analyzing these underlying text datasets (in particular CommonCrawl) [128, 129, 37, 38]. Several notable works have conducted large-scale analyses into data, particularly pretraining text corpora [124, 9, 10, 130, 131, 132, 133]. Other works have investigated the geo-diversity of vision-based datasets [52, 53, 134]. In terms of finding and visualizing datasets, a few recent tools have been proposed [135, 136].

**Web Audits**   Previous work has attempted to understand the changing landscape of the web through various web audits and to understand the evolving behavior and implications of web crawlers. A study [57] investigated how often webpages that once existed become inaccessible by looking at a sample of webpages from Common Crawl from 2013 to 2023 and found that 38% of webpages that existed in 2013 are no longer accessible a decade later. Previous work has studied the identification and characterization of traffic generated by web crawlers [59, 60, 61] including temporal analysis of web crawlers activity [62, 63]. In [64], an analysis of the top 1,000 websites in the world focused on identifying which sites are blocking popular AI web crawlers such as OpenAI's GPTBot or the Google-Extended bot and found almost 35% of these websites blocked GPTBot vs 12.5% only for the Google-Extended bot. A similar study [65] found that more than 53% of 1,164 surveyed news publishers blocked GPTBot vs 41.3% for the Google-Extended bot.

**Challenges in data transparency and its harms**   Despite efforts to document datasets [137, 138], there is a growing crisis in data transparency. The sheer scale of modern data collection and heightened scrutiny over copyright issues [139] have disincentivized thorough attribution and documentation of data lineage. This lack of transparency has led to a decline in understanding training data, as evidenced by the reduced number of datasheets [66] and the non-disclosure of training sources by prominent models [82, 140, 141]. This gap in documentation can result in data leakages between training and test sets [142, 143], exposure of personally identifiable information (PII) [144], and the perpetuation of biases and unintended behaviors [145, 146, 147], the large portions of hateful material in datasets [51, 50], all of which can cause significant harm [148]. Transparency becomes even more critical when assessing the supply chain risks of AI [149] and the ease of data poisoning in web-scale data [150].

**Data governance**   Given the importance of data documentation and auditing, and the increasing difficulty of managing and working with ever increasing datasets, various efforts have been pushed at the data governance front, including the BigScience project [151] and the Public Data Trust [152].

