# OpenReview forum: "Consent in Crisis: The Rapid Decline of the AI Data Commons"
_NeurIPS.cc/2024/Datasets_and_Benchmarks_Track — NeurIPS 2024 Track Datasets and Benchmarks Poster_

### Official Review · Reviewer_DVCm · 2024-06-28
**Review of Through the Looking-Glass: Tracing Shifts in AI Data Consent across the Web**

**Rating:** 6
**Confidence:** 5

**Review:**

The authors provide some important data and analysis that complements existing research in the field related to AI dataset collection, data use and access, and the impacts of data access and use on AI model performance, relevance to purpose, and other considerations.

Their analysis indicates that the trend will continue and increase, creating a shift away from earlier days when web scraping enabled a rich data harvest for just about any entity sending out webcrawlers. The authors conclude that the once-open doors to web data will continue to close, making it more and more difficult to readily gather web data through crawlers, and forcing more and more data behind paywalls and inside other closed environments.

Pros:
•	The research provides a valuable point-in-time snapshot based on largely-sound analysis and a substantial sample size.
•	The findings could provide helpful insights to AI developers, website owners and data controllers.
•	Although some may disagree with their conclusions and suggestions for future work, the research is a helpful and relevant contribution at a timely moment.

Cons:
•	As detailed below in this review, the authors fail to acknowledge any contextual reasons for the shift they observe and quantify. While they would not be expected to discuss or analyze those reasons in detail in this paper, failing to even allude to them does the research a disservice.
•	The state of web data accessibility illuminated by the research is a significant finding, but the paper’s warning of a dire impending dearth of data may be contingent on other factors remaining constant in a rapidly-evolving AI research field and industry with few constants. For example, the findings seem reliant on so-called general-purpose AI systems remaining as they are today, use cases remaining as-is, and the amount of training data needed to build and improve AI systems also remaining the same or growing.  For one thing, although the authors allude to this briefly in the “future of” and copyright related sections of the paper, many models customized from general-purpose AI models incorporate proprietary and subject-matter-specific datasets. It is unclear whether the current data volume demand will change as AI developers and users evolve their own needs and uses, and as developments enabling smaller AI datasets and models advance. In many ways it seems this point is an elephant in the room throughout the paper.

**Strengths:**

The paper presents solid research with several strengths:
•	The analysis provided in the paper is based on a survey incorporating a large and systemically organized sample size.
•	The survey and analysis provided by the authors illuminates an important emerging and widening chasm between the haves and have nots of web-based data access and harvesting enabled through webcrawlers. Of interest to a variety of stakeholders, their analysis indicates that the trend will continue and increase, creating a shift away from earlier days when web scraping enabled a rich data harvest for just about any entity sending out webcrawlers.
•	The authors highlight key gaps that could be helpful for website owners and publishers to understand and possibly address: 1. That the Robots.txt instructions intended to guide webcrawler data collection activities may not accurately reflect or implement appropriately the intentions detailed in website Terms of Service documentation; and 2. That some website owners block only some webcrawlers from particular AI companies, while allowing other crawlers from the same entities.

**Additional Feedback:**

The authors note that they are interested in how attitudes toward consent have evolved within a wider sample of internet domains. Future work might consider standard data “consent” instructions automated by platforms (such as automated Wordpress or Google ad related instructions), which are often left untouched by website owners, and how those decisions made by a few affect data access or restrictions for many websites.

**Clarity:**

In general, the paper provides a sufficient amount of information showing how authors arrived at their findings.

There are some minor typos discovered during this review:
•	Lines 30 and 31: incomplete sentence – “The focus of this work is to understand how increasing reliance on the internet as training data has collided with the limited protocols in place to comply with users’ intent for their content.”
•	Line 70: misspelling - “used four our study”
•	Line 273: misspelling – “One countervailing trend us”

**Correctness:**

The research findings presented in the paper are valuable, but some basis for the analysis could be improved:
•	The authors suggest that there is a notable misalignment between how ChatGPT is used, and the types of content most represented in publicly available web-based training datasets. They note that in over 30% of conversations evaluated, ChatGPT users seek outputs related to creative composition, but creative writing is ill-represented in the web data used for model training. It is not clear whether this is the most appropriate metric, since some data classified for the research using other category labels -- such as content categorized as Sexual or illegal, and Brainstorming, planning, or ideation -- could also be considered creative composition.
•	The authors refer to a misalignment between the amount of Sexual and illegal data in training corpora and the high amount of ChatGPT usage related to Sexual and illegal concepts. They acknowledge the fact that the amount of Sexual and illegal data in training corpora is relatively low because of intentional data filtering. And, the authors seem to imply that the amount of Sexual and illegal training data could better align with user interests. While this is only implied, it is a clunky argument and could be clarified.
•	The authors offer a suggestion that new machine-readable standards allowing site owners more nuanced control over scraped data uses and purposes, and better alignment with terms of service documentation should be developed. This is a worthy idea to explore, but considering the fractured standards development process related to AI and other areas of digital data, it may be an impractical suggestion.

**Documentation:**

The authors provide sufficient detail on data collection and organization, availability and maintenance, and documentation and intended uses; as well as a URL for accessing the dataset, a hosting, licensing, and maintenance plan, and sufficient detail to support reproducibility.

**Ethics:**

There are some ethical issues important to consider regarding the research:
•	The authors chose to remain anonymous which presents ethical concerns. The reader can only guess their affiliations or possible agendas that may have influenced their research. Since NeurIPS is allowing anonymity, the organization might consider requiring a detailed explanation for anonymity in relevant papers. In journalism for example, explaining the reasons for source anonymity is standard protocol.
•	When asked whether they obtained consent from people whose data they use or curate for their research, the authors state that “datasets we audit here are already publicly available for academic use.” There is no way to verify their academic affiliations, possible corporate affiliations or conflicts-of-interest.
•	The research involved an abundance of data gathering and preparation, much of which was conducted by 14 unnamed data annotation contractors. It would be valuable for research papers reflecting this sort of behind-the-scenes data labor to prominently acknowledge the people who did that work and name them as relevant contributors to the work if they desire.

While these issues are important for the paper authors and the NeurIPS community to consider, they do not warrant further ethical review according to the criteria listed in this question.

**Limitations:**

As noted above in this review, the authors state that they discuss limits of the research throughout the work, but do not include a stand-alone section condensing and highlighting those specific limits. It is unclear what the authors believe the limits of the research to be. They also fail to acknowledge any potential negative societal impacts of their work. It would improve the work were they to consider the AI development field's insistence on often-indiscriminate web data scraping, enabled by a general disregard for people's data rights and privacy. They might also consider the environmental impacts of excessive water and energy use required to produce the large general-purpose models the research seems to champion.

**Opportunities For Improvement:**

The paper could be improved were authors to address the following limitations:
•	The authors state that they discuss limits of the research throughout the work; however, a separate section condensing and highlighting those specific limits of the research would improve the work.
•	The paper’s authors lament “substantial increases in restricted tokens” erecting new blockades on a once more open web data gathering environment. But, as noted above in this review, as valuable as the survey and findings in the research are to the AI developer community, the analysis of changes to terms of service documentation and technical implementations seems utterly detached from the context within which these changes are taking place. The authors fail to recognize the reasons behind this emerging trend. Put simply, the data free-for-all that fueled the earlier days of advanced AI development is gradually being replaced with a more restrictive one; content creators, copyright holders and everyday web users posting to social sites with little control or reward are holding the reins over that data more tightly, asserting control through some of the same technical mechanisms employed by AI data collectors. The authors surely are aware of this zeitgeist, but fail to acknowledge it.

This is especially evident in the authors’ analysis of “consent” for data collection and use (a well understood and established legal concept in the areas of data privacy and human subject matter research) as something controlled by website owners, while forgetting many of the actual people who produce that data and have been affected by a lack of meaningful data consent controls for decades.

•	Although the authors provided some detail regarding the work of human annotators hired to label and classify survey data, a bit more information would enable more meaningful analysis and accountability. They merely report the mean approximate pay level for all workers -- who were located in Pakistan (8), Bangladesh (2), Vietnam (1), Philippines (1), USA (1), and Germany (1). It would have been more transparent and meaningful to report the mean pay rates for workers broken down according to each country, since pay rates likely varied.

**Relation To Prior Work:**

The authors make abundant and helpful reference to prior work related to their research.

**Summary And Contributions:**

The research surveys 14,000 web domains in an effort to assess the nature of crawlable content available on the web, particularly for the purpose of building AI training datasets. The authors analyze how content access has changed over time, and find an increase in restrictive policies that limit data capture by web crawlers, including a proliferation of new AI-specific crawler restrictions. They evaluate the effects according to various modes of web page content (text, images, audio, video) and find important discrepancies between terms of service language and actual implementation of those terms via machine readable text instructions.

---

> ### Author Rebuttal · Authors · 2024-08-17
>
> Thank you for your positive feedback, and we are encouraged that you consider our work to be a significant finding! And we do appreciate the constructive critique of our results discussion.  We very much agree with your comments, and outline the changes below that we believe will strengthen the paper accordingly.
>
> **The need for contextual discussion of these shifts**
>
> Great suggestion. We will add a subsection on this either directly into the Introduction or Discussion. Here is what we have in mind, weaving in many references to point readers toward further reading:
>
> > Context and motivations. We observe significant and rapid changes in the permissions applied to website content, which fit into and are influenced by a broader context. Most fundamentally, dataset creators and model developers may have divergent interests, including divergent financial interests, with creators in search of credit and compensation for the use of their work (Chayka, 2023) AI systems, however, may attempt to substitute for human labor (Yilmaz et al, 2023; Zarifhonarvar, 2024). Many data creators object to the use of their data in systems which may compete with them economically (Epstein et al, 2023). On top of these considerations, copyright law is unclear on several points related to AI and training data, including the authorship of the generated text (because a model is not a legal person), the boundaries of fair use, and how similar generated text has to be to something in the training corpus to be infringing (CRS, 2023; Brown, 2021).
> % Clarifying the relevant points of law will not be quick: It will take the combined work of regulatory agencies, courts and possibly national legislatures. Both creators and developers, in other words, lack the certainty about and protection for their rights that well-settled law provides.
> This atmosphere of uncertainty is compounded by the lack of adherence to a set of conventions which had gradually built up around the use of scraped or third-party data before the generative-AI era (Klassen and Fiesler, 2022; Fiesler and Bruckman, 2019), but are no longer commonly followed in AI settings (NYT v. OpenAI and Microsoft; Sancton v. OpenAI; 2023 WGA strike). These circumstances are powerful drivers of the increase in data restrictions overall: Legal and economic uncertainty lead data creators to pursue self-help.
>
> In general, our original discussion of context and societal implications were lacking (as your review points out), in part because of a hesitation to wade too far from the concrete findings, and into difficult and controversial policy topics. But we agree this context is very important, and hope that our changes above round this out in a way that is impartial and fairly represents the situation. Let us know if not!
>
> **Limitations Section**
>
> Per your recommendation, we are adding a dedicated limitations section, as well as targeted limitations in each section, so that we can contextualize each finding locally. The limitations discussions for forecasts and WildChat analysis are both shown in our general response to reviewers above.
>
> In our broader limitations section we also now incorporate your good point that “these trends are relevant to massive, data-hungry models in their current state and usage. A shift to more learning-efficient models, proprietary datasets, and/or smaller models could promise *some* solutions to invasive data extraction practices.”
>
> We would also point the reviewer to our new text in **Discussions & implications of findings** under the general response above. In the first paragraph, we round out the critical analysis of observations in a way that reflects much of your feedback. And in the second paragraph in particular, we expand on the broader social impact you recommended including.
>
> **Transparency into Annotator Pay by Country**
>
> Great suggestion! Here is the breakdown of hourly rate by country and (for comparison) each country’s hourly per capita income:
>
> Pakistan $24.05/hour ($3.13/hour)
> Bangladesh $35.92/hour ($3.85/hour)
> USA $25.25/hour ($38.98/hour)
> Philippines $25.05/hour ($5.37/hour)
>
> As you can see, our payment rate was generally far above the relevant countries’ average income and far exceeded what data annotators are normally paid (less than $2/hour in some cases: https://time.com/6247678/openai-chatgpt-kenya-workers/). (Note that while we accidentally mentioned more annotators in the previous version of the paper, only annotators from these 4 locales worked on these particular annotations.)
>
> **Ethics feedback**
>
> Thank you for these thoughtful recommendations.
>
> *Anonymity*
> Fair point – we are happy to take any steps the reviewer suggests and NeurIPS allows, including de-anonymizing. We intend in any case to include a section on competing interests in the final draft.
>
> *Consent to use data*
> On this front, we believe our conduct is in line with ethical expectations. All websites were accessed either manually or through the Wayback Machine. And when accessed through the Wayback Machine, we only “crawled” the robots.txt and Terms of Service—the two pages presumably every website would like visitors to read.
>
> *Annotator Acknowledgements*
> We appreciate this suggestion. It is important to us to recognize the work done by our annotators, without whom this study would not have been possible. We have reached out to all annotators to ask if they would like to be personally recognized in the acknowledgements of the work, and will update our draft accordingly.
>
> We hope these revisions demonstrate our commitment to engage with your feedback and strengthen our work. Please let us know if these edits reflect your expectations. If you agree they have improvedg the work, we hope that you will consider updating your scores. Thank you for your time and constructive feedback!

---

> > ### Author Response · Authors · 2024-09-01
> >
> > Dear reviewer, thank you again for your detailed feedback. We worked really hard on incorporating this feedback, and would love to hear if our responses reflect your recommendations. If you agree the changes have improved the work, we hope that you will consider updating your scores. Thank you again for your time and engagement!

---

### Official Review · Reviewer_RUtn · 2024-07-22
**Important contribution to the field**

**Rating:** 7
**Confidence:** 2
**Clarity:** The paper is clearly written.

**Review:**

This paper makes an important and timely contribution to the field. The authors claim that this is the first audit of its kind.

**Strengths:**

The paper provides an extensive review of web domains found within prominent training datasets. The paper also uses the internet archive to gain an understanding of changing attitudes over time, which is a valuable contribution.

The presented inconsistencies between robots.txt and ToS was particularly of interest.

**Additional Feedback:**

Some additional thoughts I had were:
- Given the prominence of Common Crawl as a training dataset, why was it not analyzed directly?
- Some claims could be further clarified. E.g., the authors claim that their findings show "concerning trend towards decreased openness of web data that was previously less restricted." Why is this concerning? Isn't this just an indication that website owners don't want to be crawled further (which is their right)?

**Correctness:**

There are a few slippages of terms throughout the paper. For instance, the paper claims that "In April 2024, 20.1% of the data available in a Common Crawl dump from 2019 is restricted for use in model training by robots.txt or terms of service." However, to my understanding, Common Crawl was not assessed directly. Only derivative datasets were analyzed. Thus this percentage does not take into consideration the pages that were removed from Common Crawl in the creation of C4. If that is the case, please ensure that the statements that are made are about the datasets analyzed, rather than common crawl as a whole.

**Documentation:**

The paper provides documentation regarding manual and automatic labelling and processing. The authors state that they release all annotations, though I could not find these at the time of writing this review.

**Limitations:**

This paper would benefit from a dedicated limitations section. Particularly, I am interested in the observed limitations of using the Wayback machine. There has been existing work in this area that the authors could connect this work to, e.g., Arora, Li, Youtie, and Shapira (2015).

**Opportunities For Improvement:**

The paper focuses on popular datasets, and top web domains, which may skew the results in a certain direction. For instance, might there be new websites, or smaller websites that want to be included within training corpus and use robots.txt as a means to indicate this? Further reflection on this may be beneficial.

The paper could also be extended to consider the policy implications of these findings, as well as practical recommendations to creators and users of datasets.

**Relation To Prior Work:**

This paper clearly connects with related works in this area. As mentioned, this paper could also connect to existing literature about using the wayback machine.

**Summary And Contributions:**

This paper presents an audit of the consent of web domains within large scale corpora used to train AI systems. Analysing 14,000 web domains from three major training datasets, and examining how their robots.txt files have changed over time, this paper provides insight into shifts in attitudes towards consent for being used within training data. The paper also forecasts further data restrictions over time.

The implications of this work in the discussion section were also a strength.

---

> ### Author Rebuttal · Authors · 2024-08-17
>
> We would like to thank Reviewer RUtn for thorough and constructive feedback. We are heartened that you felt this work was an important and timely contribution to the field. Below we outline several changes and answers that we believe will address your core feedback.
>
> **Crawling inclusion incentives of smaller websites/creators**
>
> This is a fantastic point, which we will add! It is certainly plausible that incentives for websites may evolve, with some sites wishing to expand their presence in AI systems, similar to SEO—especially if there is retrieval-based attribution.
>
> **Expand on policy implications & limitations**
>
> We plan to expand our discussion of findings, including limitations, in each section as well as in its own dedicated section. In particular, see the revised limitations added for forecasting and WildChat analysis, shown in the general author response above. In the methodology we have added the citation you point to, and discuss the challenges/limitations of using the Wayback Machine:
>
> > While the Wayback Machine is a valuable tool, it also poses some challenges [Arora et al, 2015], including the need to handle crawling errors, variable data formatting, duplicated data across websites, and others. The focused nature of our analysis mitigates most of these problems, however. The robots.txt files occur at a fixed relative URL (/robots.txt) and are machine-readable, terms-of-service links are generally easily recognized, and we can take advantage of a carefully prompted and spot-checked LLM to annotate the legal terms.
>
> As for recommendations to data creators and users, we’ve sharpened our narrative based on these reviews. Specifically, we focus our take-aways on the insufficiency of existing machine-readable preference signals, and the need for better standards—both to unburden creators, and enable standards-respecting developers to abide by creator preferences more accurately.
>
> **Common Crawl terminology slippages**
>
> Thank you for pointing this out—we have made these fixes throughout!
>
> **Question: Why not analyze all of Common Crawl directly?**
>
> Great question, and agree that would also be a reasonable design choice. We elected to ground the analysis more closely in the data used *directly* ingested for model training (after filtering, etc) as we felt this yields more meaningful metrics to the AI community (as opposed to a broader study of the web).
>
> **Question: Why is the rise in restrictions concerning?**
>
> As mentioned in the policy implications discussion above, we are revising this framing. We had intended to mean there are concerns for the academic researchers who respect robots.txt, caught out by restrictive preference signals that we believe are primarily intended for AI commercialization. However, this is a nuanced topic, and we now more strongly emphasize the importance of creator expression.
>
>
> We hope that our response addressed your comments and questions, and hope that we have convinced you that our paper presents a valid and important contribution which would benefit the NeurIPS Datasets community.

---

### Official Review · Reviewer_T7xg · 2024-07-23
**Interesting and extensive study on data cralwing consent in the Web**

**Rating:** 7
**Confidence:** 3
**Clarity:** Paper is well-written and clear.

**Review:**

This is a solid paper on an interesting and timely topic. A lot of data has been gathered that provides a lot of evidence for the claims made. The implications of the results are analyzed in a convincing manner. I think this paper has a lot of potential to spawn interesting discussions and follow-up research.

**Strengths:**

[S1] Very well-written paper
[S2] Interesting analyses and insights
[S3] Extensive appendix with details on the data gathering process

**Additional Feedback:**

In Section 3.1, I am confused whether the numbers really refer to percentages or percentage points. The paper stays percentage, but the referred Figure 1 rather seems to show larger increases, so it could be that the authors mean percentage points. Please clarify.

Line 87: I cannot follow the calculation that randomly sampling 10K domains from the intersection of three corpora would total in >10M domains. Please clarify.

General question (legal): Would a restriction put on crawling today have implications on the use of data for ML model training that has been crawled in the past? In privacy regulations, especially GDPR, there is the option to revoke consent on the processing of personal data. Do similar rights apply to (non-personal) data that has been crawled from a website?

There is also a related problem of using public code repositories (GitHub) for training AI models (Copilot), see https://githubcopilotlitigation.com/ - It would be interesting to extend the study to code repositories in the future, where I feel the problem is even more pressing. In case of code, there are possible infringements of open source licenses. What are the copyright licenses of general text found on, say, news websites?

**Correctness:**

Methods and study design are appropriate. I only have concerns about the predicition method, as discussed above.

**Documentation:**

Extensive details on data gathering processes are provided.

**Limitations:**

The work is limited on data from Common Crawl. However, there are other data repositories (e.g., GitHub). It would be interesting to compare to other data repositories and put the work into a broader perspective.

**Opportunities For Improvement:**

The prediction of future restrictions in robots.txt based on the SARIMA model is not motivated. Why should this be a realistic prediction, why is SARIMA a suitable model? The authors could at least validate whether using their prediction model on past developments would accurately predict the current state. (I assume this is not the case.) In the worst case, this prediction could be useless or even misleading. I would propose to either provide rationale and evidence that the model is realistic, or remove the prediction altogether from the paper.

**Relation To Prior Work:**

Yes.

**Summary And Contributions:**

In the paper, an audit of 14 000 web domains from the Common Crawl corpus is performed. These datasets are commonly used to train LLMs. The goal of the audit is to investigate the development of consent for crawling (using robots.txt as well as Terms of Services), in particular, with regards to using the content for training ML models Furthermore, the authors compare the sources of training data to the intentions when using LLM-powered chatbots.. Findings show a mismatch of specifications in robots.txt vs. Terms of Services, an increasing restriction of crawling, and a mismatch between the data sources and the use of downstream applications. The authors draw a number of interesting conclusions from their analysis.

---

> ### Author Rebuttal · Authors · 2024-08-17
>
> We would like to thank Reviewer T7xg for their encouraging comments and thoughtful feedback, which we believe will improve our work. We are delighted that the reviewer found our claims convincing, well supported, and likely to spark future research.
>
> **Forecasting**
>
> Thank you for this feedback—we agree and have de-emphasized these findings, as well as added a limitations paragraph (see the “Forecasting Limitations” paragraph in the general response). We are also adding in an analysis of historical accuracy in Appendix C.
>
> For now we have left forecasting in, with these changes, as a general indication of where trends might continue in the short run (until mid-2025) without significant external effects, but we are happy to move it to the appendix, or remove altogether if the reviewer believes this would strengthen the narrative.
>
> **GitHub future work comparison**
>
> We agree this is a very relevant question and are adding some initial analysis in our discussion of results. As a start, we can compare to the statistics on licenses and other provenance provided in “The Stack: 3 TB of permissively licensed source code”.
>
> **Additional Feedback: Section 3.1 Statistics**
>
> Thank you for pointing out these discrepancies. We will clarify them here and fix them in the paper. All reported results are actually percentages, but the per-company statistics in the text are out of date: We re-ran the analysis after obtaining more granular Wayback Machine data, and had updated the plot but not the text. We have streamlined the results text to report all the numbers consistently: not just Head_C4 results (as in the plots) but also for the Head of each dataset (C4, RefineWeb, and Dolma) and the estimated total tokens now restricted by dataset. Let us know if this clarifies your question?
>
> **Additional Feedback: Domain counts in line 88**
>
> We have clarified this sentence: “To capture this, we randomly sampled 10K domains from the intersection of the three corpora (which itself spans 10,136,147 domains).”
>
> **Legal Question: Would a restriction put on crawling today have implications on the use of data for ML model training that has been crawled in the past?**
>
> Thank you for raising this interesting question. Our legal analysis suggests that the answer to this is not clear-cut and will depend on context. In general, a change in the terms of service should not have a retroactive effect. So if a website allows crawling, an academic researcher crawls the site and publishes an analysis and the website later restricts crawling, this would not ordinarily allow the site to enforce the new terms against the researcher. However, the site may be able to successfully block future research or the dissemination of the crawled data. Moreover, it is not clear if a site’s lack of anti-AI terms in its terms or robots.txt file implies a consent to AI since this use may not have been foreseen when the terms were drafted. The lack of clarity highlights the urgent need for clearer rules in the space. We will add this point to our Discussion section.
>
> Finally, we’d like to thank you again for your positive assessment and helpful feedback. We believe these changes will strengthen the research, and hope that our response adequately addressed your comments and questions.

---

> > ### Comment · Reviewer_T7xg · 2024-08-29
> > **Thank you**
> >
> > Thank you for the clarifications and proposed changes.

---

### Official Review · Reviewer_3m6C · 2024-07-25
**An interesting investigation of how web content creators are changing the limitations they put on crawlers**

**Rating:** 8
**Confidence:** 4

**Review:**

I appreciated this paper and I'm glad that I read it. I think the need for this analysis is clear and has important policy and cultural impact.
I thought the work looking at the conflict between robots.txt and the terms of service was very powerful. There's a lot for data curators and responsible AI developers to learn from that discrepancy.

Their methods were clear with substantial detail in the appendix that I really appreciated. Both covering the human annotator training information and the technical classification details. As a small point, I felt their explanation of the different "cuts" of the data (head sample and random sample) was particularly clear - thank you for the effort you put into making that section so readable.

Where I felt this work didn't quite meet my expectations was in the synthesis and conclusions related to the findings. Is it a good or a bad thing that web content managers are starting to put restrictions on their material? The tone of the discussion seems to "warn" against data "going away" but I think there's a clear positive in the broadening education around how data on the web is used and I don't think it goes against the values of the open web to prevent crawlers from mining that text for closed / proprietary gain. In fact, the step change in robots.txt / org agent restrictions after the release of ChatGPT / GPT-4 is arguable a sign of a more healthy and better informed open web community. This also related to the finding that "Cohere, Meta, the Internet Archive, and especially Google Search receive fewer restrictions. The omissions of Cohere and Meta might be because website creators are unaware of the crawling being performed by all AI players." You don't have to agree with this interpretation, but I would have liked to see this point explored with a more critical lens.

I felt the forecast was not a strong addition to the paper. I interpreted the non-linearity in the changes over time to be yoked to releases and increased awareness of how data on the web was being used. I feel that extrapolating that timeseries has low interpretability given the impact of external events that - by definition - haven't happened yet. This point also relates to my feeling that the information hasn't been synthesised as carefully and in as much depth as it could be.

I found it interesting to see from the results that the AI models lean more heavily on news, encyclopedia and social media than on personal and organisaitonal websites and e-commerce platforms and that ChatGPT conversations tend to focus on different types of content generation. However, I'm not sure I agree with the sub-heading and conclusion that there is a "misalignment between real-world AI usage and web data". AI - even just restricting to LLMs - can be applied in many more ways than are represented in WildGPT which is just one, quite narrow, snapshot of how people are using LLMs. But mostly I found myself thinking: I don't think ChatGPT has been trained only on information from the open web.... so what exactly is the learning that the authors are looking to convey?

**Strengths:**

I think changes in availability of web data is very important to understand and this paper was clear in how they undertook the investigation. The lack of alignment between terms of service and the configuration based restrictions in robots.txt is particularly valuable.

**Additional Feedback:**

I hope I've been helpful here! I really appreciated the analyses and I think you can push yourselves to interpret them further!

A small additional point - I didn't completely understand the title? Through the looking-glass means to me that there's a parallel universe that's existing next to ours... but this analysis mostly focused on one linear timeline rather than a counterfactual?

**Clarity:**

The paper is clear, with the exceptions around the interpretations mentioned above.

**Correctness:**

With the exceptions of different potential interpretations as mentioned above, the claims made are clearly evidenced.

**Documentation:**

Yes - clear documentation - particularly in the appendix. Thank you!

**Ethics:**

No ethical concerns.

**Limitations:**

I don't feel the limitations are particularly well communicated at the moment and I've mentioned the ones I am most concerned about in earlier parts of my review. I did find the methodology clear and that is a very strong mitigation against incorrect interpretations.

**Opportunities For Improvement:**

I'd encourage you to look through the discussion and make sure that the paper results are aligned with these take home messages a little more clearly. "Empowering better protocols that respect intended consent" and "Implications for copyright and fair use" I absolutely agree with. "The future of the internet" and "the future of web-derived training data" are not well communicated at the moment. I agree that we are at risk of seeing further decreases in the open web but I don't feel the earlier parts of the paper really define the "open web" in comparison to "the web" or "walled gardens" or "content used for training AI". Isn't it a good think that we need to challenge the AI community to think more carefully about the data they are using and for what purpose?

I think you can maybe also add in a few more caveats around the forecasting or add an explanation for why you predict such a short way into the future.

**Relation To Prior Work:**

The related work section is brief but to the point. There is still significant work to do in this area and the analyses presented here take that understanding forwards.

**Summary And Contributions:**

UPDATE 2 September 2024: Score increased from 7 to 8.

-----

Quoting from the paper as I felt this summary was very clear!

> Our work has several key contributions:
>
> 1. A Large-scale audit of the web sources underlying AI training corpora. For web domains in C4, RefinedWeb, and Dolma, we trace provenance, consent mechanisms, and other factors relevant to the responsible downstream use of this data.
> 2. Proliferation of restrictions on AI training. We find a rapid proliferation of policies restricting webcrawling for AI through websites’ robots.txt and terms of use, which we use to forecast future trends. We forecast future restrictions in robots.txt and terms of use policies, estimating that 21.7% of C4 data will be restricted by mid 2025.
> 3. Consent asymmetries & inconsistencies. Consent mechanisms like robots.txt see errors and omissions in their coverage across AI companies, as well as contradictions with their terms of services—indicating inefficiencies in the tools used to signal data intentions.
> 4. Mismatch between web data and common uses of conversations AI. We quantify the large fraction of web data that comes from news, encyclopedia, forums, and social media sites, many of which monetize their content through ads and paywalls. We contrast this with real-world usage of conversational AI, showing how substantial portions of web-derived training data may be misaligned with the tasks that AI models are actually used for.

---

> ### Author Rebuttal · Authors · 2024-08-17
>
> We would like to thank Reviewer 3m6C for their constructive and positive feedback and their strong support of our submission! We are particularly encouraged that they enjoyed reading our paper, and found the conflict between robots.txt and terms of service analysis very powerful.
>
> **Synthesis and discussion of findings**
>
> Thank you for this feedback! We definitely agree the discussion of findings needs to be strengthened, and have revisited these sections (especially the Discussion paragraphs you point to). Specifically, we now more strongly emphasize the positive effects of growing awareness of data (mis)use, and expression of creator preferences. And we have worked to clarify that the rise of data restrictions is a nuanced issue, with positives, negatives, and uncertain effects. For instance, greater awareness and preference expression is good for data creators (if their preferences are enforced or providing them becomes a best practice), but may often unintentionally harm academic or non-AI researchers (who respect coarse preference indicators that may not be intended for them).
>
> Please see the proposed **Discussions & implications of findings** in the general response. We will replace most of the “future of the internet” with this, as well as some concrete analysis on how respecting creator preferences would affect data quality, heterogeneity, and freshness—a positive, not normative judgment. We believe these changes anchor the discussion in more critical analysis, and believe they are well aligned with your and other reviewers’ feedback.
>
> **Forecasting**
>
> We agree and are making a series of changes with respect to the forecasts: We are de-emphasizing these findings, adding a limitations paragraph, and pointing out that the future deeply depends on unknowable policy and legal outcomes. See the “Forecasting Limitations” section provided in the general rebuttal, which now accompanies the results.
>
> **WildChat limitations**
>
> We agree there are several limitations with WildChat and have added a limitations section (see the general rebuttal). We’ve shifted focus to what we believe are the more interesting results: the shift in the distribution of websites (and data types) available when restrictions are respected.
>
> **Defining “open web”**
>
> Thank you for pointing this out. We think the term “open web” could be ambiguous and have converted these references to “web content used to train AI models” and similar phrasing.
>
> Finally, we’d like to thank you again for your positive assessment, helpful feedback, and comments which, we believe, will vastly strengthen our paper. We hope that our response addressed your comments and questions—and let us know otherwise!

---

> > ### Comment · Reviewer_3m6C · 2024-08-31
> > **Increase score given the improved synthesis of the results**
> >
> > With my apologies for such a late reply (at the end of the summer vacation!) I acknowledge the response from the authors.
> >
> > I think the reviewers have engaged really positively with all of the reviews of their work. I really appreciate the reframing of the limitations and the forecasting analysis.
> >
> > I'm happy to increase my score from a 7 to an 8.

---

### Author Rebuttal · Authors · 2024-08-17

We thank all the reviewers for their positive and encouraging feedback!

A couple have noted how the discussion and interpretation of the findings could be improved. We have found this feedback exceptionally helpful, and propose changes below to these sections. We believe these changes improve the work, but are eager to hear how and where we might continue to improve it. We agree communicating the implications of these issues is important to get right.

A couple reviewers also advised better acknowledgement of limitations of forecasting or the WildChat analysis. Our proposed changes contextualize both sections’ analysis, and foreground discussion of limitations.

**Forecasting Limitations**

> The restriction trends are forecasted a year into the future only to provide a short-term sense of how restrictions might evolve, in the absence of significant exogenous factors. We caution the reader that this is a very strong assumption, as the outcomes of lawsuits, changing company practices, and the community's response could all have significant effects on the restrictions applied to data. These forecasts and trends are also most relevant to large, data-intensive, general-purpose models as they exist now; increased use of smaller, more specialized or less data-dependent models may reduce the relevance of the trends we identify. We accordingly focus our analysis on current statistics, and point to Appendix C for supporting information on SARIMA and backtests of our fitted model.

**Wildchat Limitations**

> Our observations come with significant caveats. The WildChat dataset may not include a representative sample of people’s interactions with language models. Not only does it include conversations only with a specific model (ChatGPT), but the WildChat proxy service is hosted on a technical website, HuggingFace Spaces, with a more technical user base that may be more likely to audit ChatGPT for inappropriate uses. Model uses change both by time and product; our analysis is specific to the model interactions collected in WildChat between April 9, 2023 to May 1, 2024, using the GPT3.5-Turbo and GPT-4 APIs. Examining how well these conclusions hold in other settings is an important subject for future work.

> Different AI products are likely to have different uses, and those uses will inevitably change over time. Finally, the use taxonomy, both for web domains and WildChat uses, was developed based with a manual, iterative process of limited granularity. Lastly, it is possible that data/information from News web domains could be used in responses for non-News classifications in WildChat, e.g. General Information. This would be exceedingly difficult to measure, and also merits analysis in future work.

**Discussions & implications of findings**

> These findings suggest websites and data creators are rapidly working to secure their data against crawling practices which often do not respect creator consent or provide attribution. Further, the use of this data for generative AI could impact the creators’ livelihoods, especially in news or the arts, as discussed in Section XX [new contextual background subsection]. While the increase in websites expressing their data-use preferences is positive, our analysis exposes several on-going challenges. First, the burden the machine-readable robots.txt standard places on these sites to enumerate all AI crawlers (without blocking other crawlers like search engines) is onerous—leaving them with significant gaps in their intended coverage. Second, our results suggest that this existing standard is insufficient for websites to express their preferences as accurately as in their terms of service. The inability to specify permissions by type of use (non-commercial, attribution, etc), rather than by individual crawler, exposes less invasive and usually more preference-respecting data uses like academic research to restrictions that may not even be intended for them. On the other hand, the lack of legal enforcement of robots.txt suggests restrictions intended for corporations may not be heeded anyway. Third, websites are not always the copyright holders, meaning existing standards may empower web-platforms at the expense of creators. Fourth, it remains ambiguous whether new restrictions can apply retroactively. Altogether, the expression of website and creator preferences is a positive trend, but it still lacks certainty around enforcement or future practices.

> Further, indiscriminate data scraping, without consent from websites or copyright holders, may not only deny credit and compensation to data creators, but also poses broader data rights and privacy concerns. Sensitive and private information, even about non-users of a website who did not share their own information with it, may become more widely available, or be cross-referenced or exposed in unanticipated situations. And there are only limited mechanisms for users to ask that their information be removed or unlearned from large models, after expensive training runs have completed.

> These practices may strongly alter the culture around consent, privacy, and sharing of information on the web more broadly. Data creators may undertake more rigorous and difficult measures to protect their data, such as anti-crawling or data-poisoning methods, or work to eliminate any online presence of their data at all. Lastly, it is worth highlighting that large-scale data collection and model pretraining are closely associated has significant environmental costs, including on water and energy usage.

Let us know if there are elements of framing and discussion that we haven’t yet included? We believe our work now better reflects the core suggestions of reviewers, and the thesis of the work: that data creators need better mechanisms to express their preferences.

The above changes and additions fit within one extra page, if the work is found to be of interest to the NeurIPS community. Thank you!

---

### Decision · Program_Chairs · 2024-09-26

**Decision:**

Accept (Poster)

**Comment:**

All reviewers note and I agree that the topic of availability of training data for AI is a timely important topic. This paper conducts a systematic and longitudinal audit of web domains and whether they are crawlable and the restrictions imposed by these sources. This is a compelling resource. The paper also attempts to do a forecast about the future restrictions of web data. This paper is timely and would make for an excellent discussion at NeurIPS. I could also see this as an oral paper potentially.